# Sinusoidal Initialization, Time for a New Start

**Alberto Fernández-Hernández**[1,†]
a.fernandez@upv.es

**Jose I. Mestre**[2,†]
jmiravet@uji.es

**Manuel F. Dolz**[2]
dolzm@uji.es

**Jose Duato**[3]
jose.duato@openchip.com

**Enrique S. Quintana-Ortí**[1]
quintana@disca.upv.es

[1]Universitat Politècnica de València
[2]Universitat Jaume I
[3]Openchip & Software Technologies S.L.

## Abstract

Initialization plays a critical role in Deep Neural Network training, directly influencing convergence, stability, and generalization. Common approaches such as Glorot and He initializations rely on randomness, which can produce uneven weight distributions across layer connections. In this paper, we introduce the *Sinusoidal* initialization, a novel deterministic method that employs sinusoidal functions to construct structured weight matrices expressly to improve the spread and balance of weights throughout the network while simultaneously fostering a more uniform, well-conditioned distribution of neuron activation states from the very first forward pass. Because *Sinusoidal* initialization begins with weights and activations that are already evenly and efficiently utilized, it delivers consistently faster convergence, greater training stability, and higher final accuracy across a wide range of models, including convolutional neural networks, vision transformers, and large language models. On average, our experiments show an increase of $4.9\%$ in final validation accuracy and $20.9\%$ in convergence speed. By replacing randomness with structure, this initialization provides a stronger and more reliable foundation for Deep Learning systems.

## 1 Introduction

Weight initialization remains a critical yet often underappreciated component in the successful training of Deep Neural Networks (DNNs) (including Convolutional Neural Networks (CNNs) and Transformer-based architectures). Although numerous training strategies and architectural innovations receive substantial attention, initialization itself frequently remains in the background, implicitly accepted as a solved or trivial problem. Nonetheless, the initial setting of weights profoundly affects training dynamics, influencing convergence speed, stability, and even the final model's ability to generalize [LeCun et al., 1998, Glorot and Bengio, 2010, He et al., 2015b].

Historically, weight initialization has evolved significantly. Early methods such as those proposed by LeCun et al. [1998] were primarily heuristic, offering basic variance scaling principles. The seminal work of Glorot and Bengio [2010] rigorously connected weight initialization to variance preservation across layers, introducing a randomized scaling approach that became foundational for Deep Learning (DL). Later, He et al. [2015b] refined this idea, tailoring variance preservation specifically to networks with Rectified Linear Units (ReLUs), and their approach quickly established itself as a default choice in modern DNN training pipelines.

---

[†]These authors contributed equally to this work.

39th Conference on Neural Information Processing Systems (NeurIPS 2025).

Despite the established efficacy of these stochastic variance-preserving methods, a fundamental question remains largely unchallenged in current literature: *Is randomness fundamentally necessary for effective neural network initialization?* Random initialization introduces variability that can complicate reproducibility, debugging, and systematic experimentation. Moreover, it implicitly assumes that the optimal starting conditions for training must inherently involve stochasticity, without thoroughly exploring deterministic alternatives that might offer similar or even superior statistical properties.

In this paper, we directly address this scarcely explored assumption by proposing a novel deterministic initialization scheme, termed *Sinusoidal* initialization. This technique preserves the critical variance-scaling property central to the effectiveness of Glorot and He initialization schemes, while completely removing randomness from the initialization procedure. It achieves this by employing *Sinusoidal* functions to deterministically initialize weight matrices, ensuring that each neuron in a given layer receives a distinct weight configuration. This use of *Sinusoidal* patterns enables rich diversity in weight values while maintaining global variance characteristics, striking a balance between structure and expressive power.

Specifically, our contributions are threefold:

- We introduce the *Sinusoidal* initialization, a deterministic initialization strategy utilizing *Sinusoidal* patterns, explicitly emphasizing symmetry, functional independence, and rigorous variance control for signal propagation, thus eliminating stochastic uncertainties.

- We present a sound theoretical framework that reveals how stochastic initialization schemes can hinder neuron activation efficiency by inducing asymmetries and imbalances in neuron behavior. By contrast, deterministic approaches that enforce structural symmetry, such as the proposed *Sinusoidal* initialization, naturally mitigate these issues, leading to more stable activations and improved gradient propagation across layers.

- Through extensive empirical evaluations involving CNNs, including some efficient architectures, Vision Transformers, and language models, we rigorously demonstrate the empirical superiority of the *Sinusoidal* initialization over standard stochastic initializations, demonstrating faster convergence and higher final accuracy.

The *Sinusoidal* initialization consistently yields higher final accuracy and faster convergence, as detailed in Section 4. These benefits are already evident in the case of ResNet-50 trained on CIFAR-100, presented in Figure 1 as a representative example. The experimental setup and results for additional DNN architectures are also provided in Section 4.

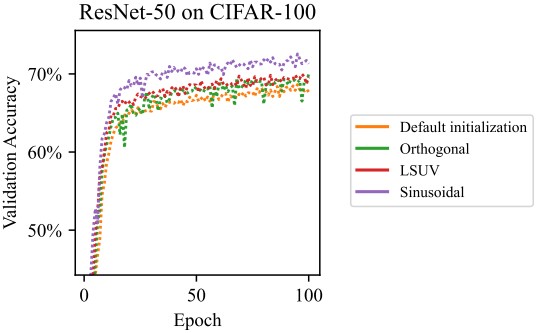

Figure 1: Validation accuracy over training epochs for ResNet-50 on CIFAR-100, comparing *Sinusoidal* with other initialization schemes.

By challenging the ingrained belief in randomness as a necessary condition for initialization, we encourage new avenues for rethinking foundational aspects of DNN training, advocating for deterministic and comprehensive practices in the future of DL research.

The remainder of this paper is organized as follows. We begin by revisiting the foundations of weight initialization and its role in DL in Section 2. We then introduce the proposed initialization scheme, outlining its definition and theoretical motivations along Section 3. This is followed by an extensive empirical evaluation in Section 4 across diverse architectures and tasks. Finally, we conclude by summarizing our findings and outlining future research directions in Section 5.

## 2 Background: Revisiting Initialization

Effective weight initialization is deeply rooted in the principles of variance preservation, which aim to maintain stable signal propagation across layers. This stability is crucial to avoid the notorious problem of vanishing or exploding gradients, which hinder DNN training [Hochreiter et al., 2001].

Formally, for a layer with weight matrix $W \in \mathbb{R}^{m \times n}$, Glorot and Bengio [2010] proposed Glorot initialization, where each weight is sampled from a distribution (either uniform or normal) with zero mean and variance $\text{Var}(W) = 2/(m + n)$, specifically chosen to balance the flow of information during both the forward and backward passes.

Subsequently, Glorot et al. [2011] observed that ReLUs produce naturally sparse activations, typically leaving roughly $50\%$ of hidden units active (positive) and the other $50\%$ inactive (zero). They highlighted the significance of this balanced activation pattern for efficient gradient flow and effective learning. Recognizing this characteristic, He et al. [2015b] refined this initialization specifically for networks using ReLU activations, proposing the variance $\text{Var}(W) = 2/n$ which analytically accounted for ensuring robust data propagation.

While these stochastic methods have been widely successful, their intrinsic randomness introduces significant variability. In particular, the random sampling procedure can sometimes produce weight matrices that lead to suboptimal neuron configurations in the hidden layers. This can cause slow convergence, instability, or even divergence early in training.

Various methods have attempted to reduce or structure this randomness. Orthogonal initialization [Saxe et al., 2014], for instance, ensures orthogonality of weight matrices, stabilizing signal propagation and improving gradient norms, but still involves random orthogonal matrix sampling. Layer-Sequential Unit-Variance (LSUV) [Mishkin and Matas, 2016] takes a step further by adjusting randomly initialized weights iteratively until achieving a precise activation variance. However, LSUV remains data-dependent and partially stochastic, limiting direct reproducibility and practical convenience.

Building on this line of work, some recent approaches have explored fully deterministic alternatives to random initialization. Blumenfeld et al. [2020] state that DNNs can be trained from nearly all-zero, symmetric initializations, as long as some form of symmetry-breaking, such as dropout or hardware-level non-determinism, is present. Zhao et al. [2022] propose a scheme based on identity and Hadamard transforms, which achieves benchmark performance on ResNet architectures while preserving signal propagation. While both methods demonstrate that randomness is not strictly necessary for successful training, they primarily aim to match the performance of standard random initializations in terms of final accuracy. Crucially, they do not report improvements in convergence speed, robustness across architectures, or theoretical insights into the limitations of randomness. These works play an important role in establishing deterministic initialization as a viable alternative, but remain early steps in the broader effort to understand and fully exploit its potential.

A notable gap thus remains in the literature: Despite these advances, there is currently no widely adopted deterministic initialization method with demonstrated potential to surpass the performance of random initializations. This scarcely explored direction motivates our work, wherein we propose a fully deterministic alternative, explicitly designed to deliver improved accuracy and convergence speed across a wide range of architectures.

## 3 Definition and Theoretical Foundation of Sinusoidal Initialization

In this section, we introduce a novel deterministic initialization scheme for DNNs, which we refer to as *Sinusoidal* initialization. This approach is proposed as a theoretically-motivated alternative to conventional stochastic methods, aiming to enable efficient signal propagation and promote functional independence among neurons from the very first training step.

### 3.1 Sinusoidal Initialization

Let $W \in \mathbb{R}^{m \times n}$ denote the weight matrix of a neural network layer, where $n$ represents the number of input features and $m$ the number of output neurons. We define the weights deterministically using

sinusoidal functions as

$$W[i, j] = a \cdot \sin(k_i \cdot x_j + \phi_i),  \tag{1}$$

where

$$k_i = 2\pi i, \ x_j = j/n \ \text{ and } \ \phi_i = 2\pi i/m,$$

for $i \in \{1, 2, \ldots, m\}$ and $j \in \{1, 2, \ldots, n\}$.

Under this formulation, each row $W[i, :]$ corresponds to a uniformly sampled sinusoidal waveform of frequency $k_i$ and phase offset $\phi_i$, evaluated over an evenly spaced grid of $n$ points in the interval $[0, 1]$. The linear increase in frequency with the row index ensures that the function $f_i(x) = a \cdot \sin(k_i \cdot x + \phi_i)$ completes exactly $i$ full oscillations. The phase offsets $\phi_i$ are also chosen to be equally spaced to avoid identical first values across the rows of matrix $W$.

The amplitude $a$ is chosen according to theoretical foundations established by Glorot and Bengio [2010] and He et al. [2015b], which dictate the variance of weights in order to enable efficient signal propagation. Specifically, in our particular case, we adhere to the variance proposed by Glorot, setting the variance of $W$ to be $2/(m + n)$. If $v$ denotes the variance of $W$ when $a = 1$, it then follows that $2/(m + n) = a^2 v$, which yields a natural way to determine the amplitude $a$.

Furthermore, we follow standard practice by initializing all neuron biases to zero, ensuring no preactivation offset is introduced at the start of training.

**Weight Distribution**

Unlike traditional random initializations, this *Sinusoidal* scheme yields a non-Gaussian distribution of weights. As illustrated in Figure 2, the weights follow an arcsine distribution, exhibiting higher density near the extrema $\pm a$ and lower density near zero. This contrasts with the symmetric bell-shaped curves centered at zero typically produced by standard stochastic initialization methods. Additionally, Figure 3 provides a heatmap of matrix $W$, showcasing the underlying harmonic structure induced by the *Sinusoidal* construction. The structured and oscillatory pattern of each row is reminiscent of sinusoidal positional encodings[1] employed in Transformer architectures [Vaswani et al., 2017], thereby offering spatial diversity without relying on randomness.

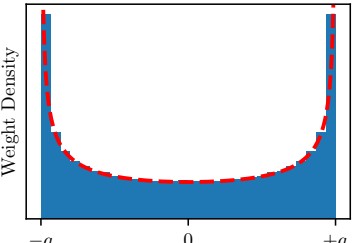

Figure 2: Weight distribution under *Sinusoidal* initialization.

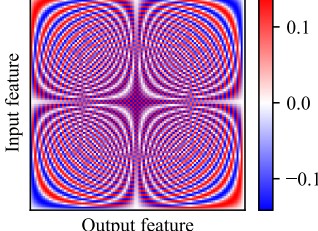

Figure 3: Heatmap of the initialized weight matrix.

This initialization strategy provides a well-conditioned starting point for early training stages, as will be further discussed in the next subsection.

## 3.2 Theoretical Motivation

This section provides a theoretical motivation for the design of the proposed initialization. We emphasize its ability to foster efficient training dynamics from the earliest stages. In particular, we analyze the impact that initialization induces on neuron activation patterns in the intermediate layers of DNNs. Furthermore, we introduce a quantitative framework to assess activation quality.

We develop the following theoretical analysis for DNNs employing *ReLU-like* activations, which are functions such as Gaussian Error Linear Unit (GELU), Sigmoid Linear Unit (SiLU), or Parametric

---

[1]Positional Encodings were indeed the initial inspiration behind our initialization. The reader may note some differences, such as the use of phase shifts in the waves, the exclusive use of sine (as opposed to both sine and cosine), and the adoption of increasing rather than decreasing frequencies. Nonetheless, the underlying connection lies in the expressive power of harmonic functions to assign independent vectors, a fact long understood since Fourier.

Rectified Linear Unit (PReLU), whose shapes resemble the classical rectified activation. In such networks, we consider a neuron to be *active* if its output is positive, and *inactive* otherwise. Unlike ReLU, these activations may still propagate small negative values in the inactive state, resulting in a smoother transition between activation regimes.

Intuitively, a neuron is inefficient when its activation pattern is imbalanced—*i.e.*, when it remains either active or inactive for a disproportionate amount of samples. Such behavior severely limits its representational power. We formalize this notion through the concept of a *skewed neuron*:

**Definition 1.** Let $Z$ be a random variable representing the output of a neuron with corresponding weights $W_1, W_2 \ldots, W_n$, and let $\alpha \in (0, 1/2)$. The neuron associated with $Z$ is said to be *skewed with degree $\alpha$* if

$$|P(Z > 0 \mid W_1, W_2, \ldots, W_n) - 1/2| > \alpha.$$

This definition quantifies the deviation of a neuron's activation from the ideal $50\% - 50\%$ balance. For instance, with $\alpha = 0.3$, a neuron is considered *skewed* if the probability difference between activation and inactivation in absolute value exceeds the $80\% - 20\%$ threshold.

A high prevalence of *skewed* neurons impairs training efficiency by reducing the network's capacity to exploit nonlinearities and undermining its expressive power, and hence impeding optimization and slowing down convergence.

**Empirical evaluation of activation imbalance**

To analyze how different initialization schemes affect activation balance, we conduct an experiment with the ViT B16 on the ImageNet-1k dataset. The goal is to compare the extent of activation imbalance across several common initialization strategies, such as *Glorot*, *He*, *Orthogonal*, *LSUV*, and our proposed *Sinusoidal* initialization. Figure 4 visually illustrates neuron activation states of the last Feed-Forward of the ViT. Each plot maps neuron indices (horizontal axis) against 768 randomly selected input samples (vertical axis), with white pixels indicating active (positive output) neurons and black pixels indicating inactive (negative output) ones.

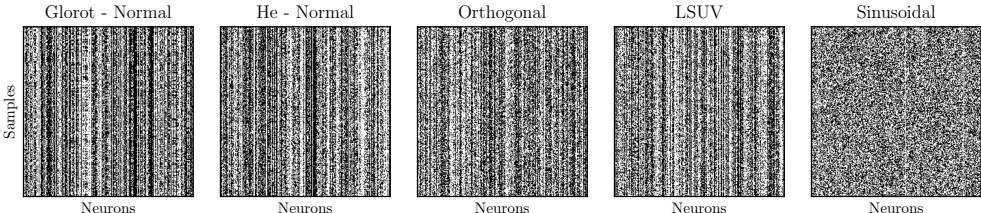

Figure 4: Neuron activation states, in white active neurons ($> 0$), for different initializations.

Traditional initializations exhibit distinct vertical white or black bands, indicating neurons that remain consistently active or inactive—*i.e.*, *skewed* neurons. As noted in Section 2, Glorot et al. [2011] emphasized the importance of balanced ReLU activations, with roughly $50\%$ of neurons active to ensure effective learning. *Skewed* neurons disrupt this balance, limiting the network's expressive power by underutilizing its nonlinearity and pushing it toward more linear behavior. In contrast, the *Sinusoidal* initialization yields a more irregular, noise-like pattern with no dominant columns, suggesting a more balanced and expressive activation regime.

To complement this visual analysis, we quantified the proportion of *skewed* neurons under two thresholds for $\alpha$: 0.1 and 0.3. As shown in Table 1, conventional initialization schemes result in an unexpectedly high proportion of *skewed* neurons, whereas the *Sinusoidal* approach barely exhibits any *skewed* neuron. These results highlight the potential of *Sinusoidal* initialization to better preserve activation diversity. For further experimental details, refer to Appendix C.2.

Table 1: Percentage of *skewed* neurons under different $\alpha$ thresholds and initialization schemes.

| $\alpha$ | **Glorot** | **He** | **Orthogonal** | **LSUV** | **Sinusoidal** |
|---|---|---|---|---|---|
| 0.1 | 82.7 % | 82.8 % | 84.0 % | 84.3 % | **0.2 %** |
| 0.3 | 51.9 % | 51.4 % | 49.6 % | 50.9 % | **0.2 %** |

These findings reveal a strikingly high proportion of *skewed* neurons under conventional random initializations, whereas the proposed deterministic *Sinusoidal* initialization yields a significantly lower rate. This contrast naturally raises a critical question: *What is the reason for this phenomenon?* The following subsection aims to shed light on the inner workings of random initializations, why they tend to produce *skewed* neurons, and why the *Sinusoidal* scheme successfully mitigates this issue.

**Impact of randomness**

It is traditionally assumed that randomness in weight initialization is essential for effective training in DNNs. However, the results presented in this work challenge this long-standing belief, revealing that randomness may in fact seed unfavorable configurations from the very beginning.

In a standard random initialization, the weights $W_1, W_2, \ldots, W_n$ of a neuron are independently sampled from a distribution centered at zero. We identify a critical statistic that governs the initial behavior of the neuron: the cumulative weight imbalance, quantified by the sum

$$S = W_1 + W_2 + \cdots + W_n.$$

This quantity captures the imbalance between positive and negative weights. Despite the zero-mean assumption, the value of $S$ can deviate substantially from zero due to statistical fluctuation. For example, under the He initialization [He et al., 2015b], where weights are generated with variance $\sigma^2 = 2/n$, the Central Limit Theorem implies that $S \sim \mathcal{N}(0, 2)$.

To empirically investigate the relationship between this statistic and the emergence of *skewed* neurons, we take, for simplicity, a Multilayer Perceptron (MLP) as the reference model on which to conduct our experiments. For full experimental details, see Appendix C.2. When focusing solely on the first layer of the network, we observe a crystal-clear correlation between the two concepts.

To illustrate this relationship, Figure 5 presents a histogram of $S$ values across all neurons in the first hidden layer. Blue bars correspond to neurons classified as *skewed* at level $\alpha = 0.3$ (*i.e.* with *skewed* balance greater than $80\% - 20\%$), while orange bars denote non-*skewed* neurons. The plot reveals a striking pattern: neurons with large absolute values of $S$ are invariably *skewed*, confirming a strong empirical correlation between $|S|$ and the degree of *skewness*. This reinforces the central role of $|S|$ in determining activation imbalances at initialization.

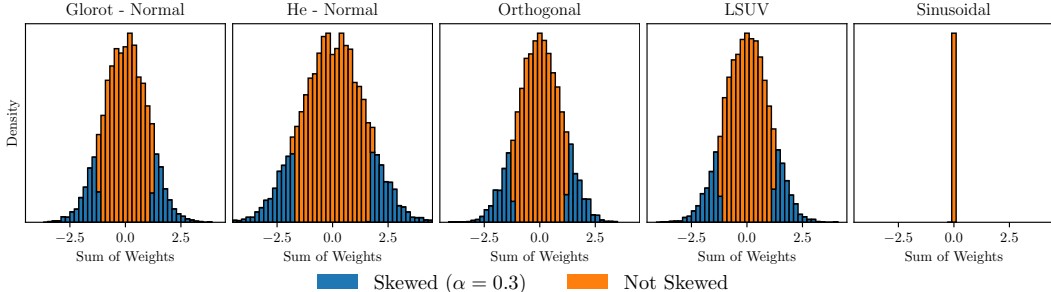

Figure 5: Histogram of the statistic $S$ values, highlighting the link between large $|S|$ and neuron *skewness* at $\alpha = 0.3$.

This observation makes it evident that, at the first layer of the model, the relationship between *skewed* neurons and the corresponding value of the statistic $S$ is remarkably precise: for each value of $\alpha$, there exists a unique threshold $\lambda$ such that a neuron is *skewed* of degree $\alpha$ if and only if $|S| > \lambda$. This equivalence can be formally established, as stated in the following theorem.

**Theorem 1** (Threshold Equivalence). *Let $W_1, W_2, \ldots, W_n$ and $X_1, X_2, \ldots, X_n$ be i.i.d. sequences with*

$$\mathbb{E}[W_1] = 0, \quad \mathbb{E}[X_1] = \mu > 0, \quad \mathrm{Var}(W_1) = \theta^2 \in (0, \infty), \quad \mathrm{Var}(X_1) = \sigma^2 < \infty.$$

*Define $S_n = \sum_{i=1}^{n} W_i$ and $Z_n = \sum_{i=1}^{n} W_i X_i$. Then for any $\alpha \in (0, 1/2)$, define*

$$\lambda_n(\alpha) := \left( \theta \sigma \sqrt{n} \right) / \mu \cdot \Phi^{-1} \left( 1/2 + \alpha \right),$$

*where $\Phi^{-1}$ is the quantile function of the standard normal distribution. Then, as $n \to \infty$, the following equivalence holds with probability tending to one:*

$$|P(Z_n > 0 \mid W_1, W_2, \ldots, W_n) - 1/2| > \alpha \quad \text{if and only if} \quad |S_n| > \lambda_n(\alpha).$$

This theoretical result establishes a tight connection between the statistic $S$ and the *skewness* property of neurons. As a direct consequence, stochastic initializations obtained by sampling from any zero-mean distribution inevitably lead to skewed neurons with probability tending to one. Formally:

**Theorem 2** (Asymptotic skewness under random initialization). *Consider a neuron with zero bias and input $X_1, \ldots, X_n$ and weights $W_1, \ldots, W_n$. Assume $\{W_i\}$ are i.i.d. with $\mathbb{E}[W_1] = 0$ and $\mathrm{Var}(W_1) = \theta^2 \in (0, \infty)$, and $\{X_i\}$ are i.i.d., independent of $\{W_i\}$, with $\mathbb{E}[X_1] = \mu > 0$ and $\mathrm{Var}(X_1) = \sigma^2 < \infty$. Let $Z_n = \sum_{i=1}^{n} W_i X_i$. Then for every deterministic sequence $\alpha_n \downarrow 0$, the probability of having*

$$\left| P(Z_n > 0 \mid W_1, \ldots, W_n) - 1/2 \right| > \alpha_n$$

*tends to one as $n \to \infty$. Equivalently, with probability tending to one there exists a nonzero skewness level at which the neuron is skewed.*

Therefore, in order to design initializations that do not systematically induce skewed neurons, it is necessary to move beyond the classical family of random schemes that fall under the assumptions of Theorem 2. This motivates the exploration of deterministic initializations specifically crafted to enforce balance from the outset. In the particular case of the *Sinusoidal* initialization, the connection with Theorem 1 becomes especially meaningful. Unlike standard random initialization schemes which offer no guarantees on the aggregate behavior of individual neurons, the *Sinusoidal* initialization exhibits an inherent structural regularity. This symmetry manifests in the exact cancellation of weights along each row of the initialization matrix, ensuring that each neuron begins with a perfectly balanced contribution across its inputs. The following result makes this property precise:

**Theorem 3** (Row Cancellation). *Let $W \in \mathbb{R}^{m \times n}$ be defined as in Equation* (1) *with $m < n$. Then, for every row index $i \in \{1, 2, \ldots, m\}$, the entries along that row sum to zero:*

$$\sum_{j=1}^{n} W[i, j] = 0.$$

The proofs and further theoretical analysis can be found in Appendix B.

Consequently, if the objective is to initialize the weights without introducing any biased (skewed) neurons, then, at least in the first layer, we should aim to enforce $S = 0$ for every neuron.

For deeper layers though, the correlation between neuron *skewness* and the magnitude of $S$ progressively fades due to the asymmetries introduced by preceding layers in the established stochastic initializations. Indeed, the input to an intermediate layer is already biased, and the mean of each input component can deviate more and more, gradually decoupling the notions of *skewness* and $|S|$. This issue, however, is naturally mitigated by the *Sinusoidal* initialization. Owing to its strong symmetry, which ensures $S = 0$, it guarantees that each neuron begins with an input that is perfectly centered, thereby eliminating any intrinsic directional bias. As a result, the *Sinusoidal* initialization is theoretically guaranteed to prevent the emergence of *skewed* neurons, an undesirable behavior frequently observed under stochastic schemes. This conclusion is not only supported by the theorems presented above but also empirically validated in Table 1, where the proportion of *skewed* neurons is identically zero across all thresholds tested.

The relevance of this symmetry during training will be quantitatively assessed in Section 4. However, before addressing training dynamics, it is crucial to examine a second foundational property: whether individual neurons are initialized with sufficiently diverse and independent functional responses.

**Functional independence across neurons**

Another critical consideration is the functional independence among neurons in a layer. An optimal initialization scheme should ensure that each neuron focuses on distinct aspects of the input, thereby enhancing representational efficiency.

To detect potential correlations among neurons, we examine horizontal patterns in Figure 4. A dark or white horizontal band would indicate that all neurons respond similarly to a specific input sample, signaling redundancy. As observed, none of the schemes, including the *Sinusoidal* one, exhibit dominant horizontal patterns. In our case, functional independence is enforced through distinct frequencies $k_i$ and phase offsets $\phi_i$ for each neuron, evoking the orthogonality of Fourier bases.

In order to further look into more intrinsic horizontal dependencies, the Overfitting–Underfitting Indicator (OUI), as introduced by the authors in Fernández-Hernández et al. [2025], can be employed to numerically assess functional independence across neurons. This indicator measures the similarity between activation patterns of sample pairs within a given layer, assigning values in the range $[0, 1]$, where 0 indicates highly correlated activations and 1 denotes complete independence. In this context, the activation pattern of a sample is understood as a binary vector, where each entry corresponds to the on/off activation state of a hidden neuron. Hence, OUI quantifies how differently pairs of samples activate the network, providing a direct measure of expressive capacity at initialization.

While all methods yield reasonably high OUI scores—Glorot (0.56), He (0.59), Orthogonal (0.57), and LSUV (0.58)—the *Sinusoidal* initialization consistently achieves the highest value, reaching an outstanding 0.98. This indicates that, although traditional initializations already promote a significant degree of activation diversity, the *Sinusoidal* scheme provides a strictly better starting point with respect to this metric, offering a more expressive and balanced activation regime from the very beginning.

**Theoretical conclusions**

The findings of this section lead to two key insights regarding initialization strategies for DNNs:

1. Random initializations naturally induce a high proportion of *skewed* neurons, which can compromise training efficiency during early stages.

2. Deterministic initializations can substantially mitigate this issue, provided they are designed with a structure that incorporates symmetry, functional independence, and variance control.

From this perspective, the proposed *Sinusoidal* initialization emerges as a strong candidate: it combines a bounded function family with inherent symmetry, sufficient expressiveness to ensure neuronal independence, and an analytically tractable formulation. Together, these properties make it a promising alternative for establishing a new standard in the initialization of DNNs.

## 4    Experimental Benchmarking of Sinusoidal initialization

Having introduced the *Sinusoidal* initialization, a novel deterministic scheme theoretically motivated, we now turn to its empirical validation against both classical and state-of-the-art (SOTA) initialization methods across a range of DNN configurations.

Our experimental validation specifically addresses two main hypotheses: 1) **Accuracy improvement**: The *Sinusoidal* initialization achieves higher final validation accuracy compared to alternative initialization schemes in a significant proportion of cases. 2) **Accelerated convergence**: The *Sinusoidal* initialization converges faster than any other initialization considered, as measured by the Area Under the Curve (AUC) metric, which besides for classification evaluation, can be used to compare the training convergence speeds [Huang and Ling, 2005].

To evaluate these hypotheses rigorously, we conducted experiments involving five diverse configurations of DNNs and datasets: ResNet-50 [He et al., 2015a] trained on CIFAR-100 Krizhevsky et al. [2009], MobileNetV3 [Howard et al., 2019] trained on Tiny-ImageNet Le and Yang [2015], EfficientNetV2 [Tan and Le, 2021] trained on Tiny-ImageNet, ViT-B16 [Dosovitskiy et al., 2021] trained on ImageNet-1K Russakovsky et al. [2015]), and BERT-mini [Devlin et al., 2019] trained on WikiText Merity et al. [2016]. For the first three configurations, we utilized SGD, Adam, and AdamW optimizers, while ViT-B16 and BERT-mini were trained exclusively with SGD and AdamW, respectively, due to the inability of other optimizers to achieve effective training. To ensure a fair comparison between training processes, learning rate schedulers were not used, which may affect the comparability of final accuracies with benchmarks focused on maximizing accuracy.

We trained each configuration fully from scratch using four initialization methods: Default (variants of the He initialization detailed in Table 2), Orthogonal (Orth.), LSUV, and our proposed *Sinusoidal* initialization (Sin.). The exact training procedures, including hyperparameter settings and implementation details, are provided in Appendix C.3. We measured the maximum validation accuracy achieved at epoch 1, epoch 10, and overall, as well as the AUC for every training scenario.

Table 2: Default initializations for each model.

| Model | Layer type | Default initialization | Activation type |
|---|---|---|---|
| ResNet-50 | Conv2D Linear | He normal He uniform | ReLU |
| MobileNetV3 | Conv2D Linear | He normal Normal | HardSwish |
| EfficientNetV2 | Conv2D Linear | He normal Uniform | SiLU |
| ViT-B16 | Conv2D Linear | Truncated normal Glorot uniform | ReLU GeLU |
| BERT-mini | Linear | Normal | GeLU |

In all our setups, to fairly compare initializations using AUC across different runs, we ensure that all models are trained for the same number of epochs. For each combination of model, dataset, and optimizer, training is performed with early stopping to guarantee convergence for every initialization. Once the slowest run has converged, the remaining models continue training for the same number of additional epochs until reaching the maximum epoch count. This procedure ensures that (1) all trainings reach their maximum validation accuracy, and (2) the number of epochs used to calculate the AUC for measuring convergence speed is consistent across initializations. The detailed experimental outcomes are summarized in Table 3 and illustrated in Figure 6.

Table 3: Validation accuracy (%) after 1 epoch, 10 epochs, and maximum accuracy, as well the AUC for each optimizer and initialization scheme (Default, Orthogonal, LSUV, *Sinusoidal*).

| Model / Dataset | Optim. | 1 epoch accuracy (%) | | | | 10 epoch accuracy (%) | | | | Maximum accuracy (%) | | | | AUC | | | |
|---|---|---|---|---|---|---|---|---|---|---|---|---|---|---|---|---|---|
| | | Def. | Orth. | LSUV | Sin. | Def. | Orth. | LSUV | Sin. | Def. | Orth. | LSUV | Sin. | Def. | Orth. | LSUV | Sin. |
| ResNet-50 CIFAR-100 | SGD | 2.6 | 4.0 | 3.6 | **5.0** | 9.7 | 18.2 | 14.8 | **24.3** | 37.3 | 46.5 | 44.8 | **51.9** | 25 | 35 | 31 | **42** |
| | Adam | 10.1 | 12.5 | 11.8 | **19.9** | 39.9 | 44.1 | 43.1 | **48.7** | 53.1 | 56.6 | 56.3 | **61.5** | 48 | 51 | 51 | **57** |
| | AdamW | 12.9 | 13.9 | 12.6 | **23.1** | 58.5 | 60.9 | 62.5 | **63.7** | 67.5 | 67.7 | 69.7 | **71.0** | 64 | 65 | 66 | **68** |
| MobileNetV3 TinyImageNet | SGD | 0.7 | 0.8 | **1.4** | 1.2 | 1.5 | 3.0 | 2.9 | **4.5** | 18.4 | 25.8 | **28.0** | 21.6 | 26 | **38** | 35 | 36 |
| | Adam | 5.0 | **8.4** | 5.4 | 5.5 | 22.0 | 24.8 | **26.7** | 24.0 | 32.8 | 34.4 | **35.2** | 34.8 | 62 | 66 | **71** | 65 |
| | AdamW | 13.4 | 14.5 | **14.5** | 12.3 | 40.9 | **43.2** | 40.1 | 42.1 | 40.9 | **43.6** | 40.1 | 42.6 | 79 | **85** | 76 | 82 |
| EfficientNetV2 TinyImageNet | SGD | 1.0 | 1.2 | **1.9** | 1.0 | 5.0 | 5.3 | **13.4** | 9.0 | 28.1 | 30.9 | **32.1** | 32.0 | 47 | 52 | **56** | **56** |
| | Adam | 2.9 | 4.2 | 5.6 | **7.2** | 19.3 | 20.2 | 23.0 | **26.4** | 27.7 | 29.8 | 32.7 | **36.6** | 53 | 56 | 62 | **70** |
| | AdamW | 9.6 | 9.3 | **11.3** | 9.4 | 48.1 | 48.2 | 48.7 | **51.0** | 50.0 | 50.2 | 49.3 | **53.5** | 100 | 100 | 100 | **106** |
| ViT-16 ImageNet-1k | SGD | 2.8 | **3.8** | 3.5 | 2.6 | 15.2 | **15.2** | 14.1 | 13.9 | 28.6 | 28.2 | 29.6 | **31.5** | 23 | **25** | 24 | **25** |
| BERT-mini WikiText | AdamW | 9.4 | 6.3 | **11.0** | 6.3 | 13.6 | 10.9 | 14.8 | **17.0** | 40.4 | **42.2** | 15.9 | 41.1 | 58 | **72** | 32 | **72** |

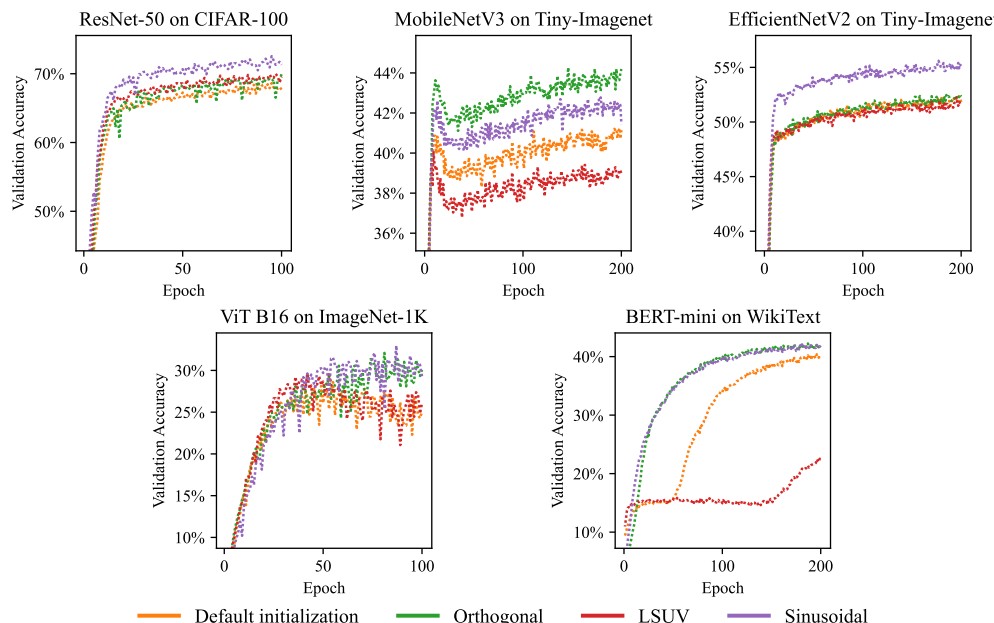

Figure 6: Training curves showing that *Sinusoidal* initialization leads to faster convergence compared to default, orthogonal, and LSUV initializations across architectures, using the optimizer that achieved the highest accuracy (SGD for ViT, AdamW for the others).

Analyzing these tables, several clear trends emerge. Firstly, our *Sinusoidal* initialization consistently achieves higher final validation accuracy compared to other initializations, supporting our first hypothesis. Notably, in the case of MobileNetV3 on TinyImageNet, Orthogonal initialization slightly outperforms our method. Despite this isolated exception, the general trend strongly favors our *Sinusoidal* initialization. This particular case, however, highlights that this research field remains rich with open avenues for exploration. We conducted an additional experiment replacing the HardSwish activation in MobileNetV3 with a ReLU, and under this setting, Sinusoidal once again outperformed all other initialization strategies, consistent with the rest of our results. This suggests that the observed behavior stems from the interaction between the HardSwish activation and the initialization scheme. This particular case, however, highlights that this research field remains rich with open avenues for exploration. The behavior observed in this architecture offers a valuable opportunity to further advance our understanding of the DNN training process and refine *Sinusoidal* initialization strategy accordingly.

Secondly, regarding convergence speed, the *Sinusoidal* initialization outperforms every other initialization in AUC, except in MobileNet, which indicates a faster convergence and higher accuracy in average over the training process. It also consistently surpasses all other methods in terms of validation accuracy at early stages of training, specifically at epochs 1 and 10, unequivocally confirming our second hypothesis.

Integrating these observations into high-level metrics, our experiments collectively yield an average increase in final validation accuracy of $4.9\%$ over the default initialization, $1.7\%$ over orthogonal and $3.4\%$ over LSUV. Considering the overall training process, the AUC indicates an average improvement of $20.9\%$ over the default initialization, $5.7\%$ over orthogonal and $18.0\%$ over LSUV.

These results strongly support the theoretical predictions from the previous section and underline the practical advantages of the proposed *Sinusoidal* initialization. Such significant and consistent gains in accuracy and convergence speed emphasize the potential of deterministic, symmetry-driven initialization methods as effective tools for enhancing the performance and efficiency of DNN training.

## 5 Conclusion

In this article, we introduced and extensively studied the *Sinusoidal* initialization, a novel deterministic approach to initializing DNNs. Our method fundamentally departs from the traditional paradigm of stochastic initializations, employing structured *Sinusoidal* patterns designed to enforce symmetry, functional independence, and precise variance control from the outset.

Through rigorous theoretical analysis, we demonstrate that conventional random initialization schemes inherently introduce asymmetries and imbalances, resulting in a high prevalence of what we define as *skewed* neurons, a notion we formalize and quantify in this work. This *skewness* substantially undermines neuron activation efficiency and may restrict the expressiveness of the network. In contrast, the deterministic structure of the *Sinusoidal* initialization is theoretically proven to inherently mitigate these effects.

Empirical evidence strongly supports our theoretical claims. Extensive experiments across diverse DNN architectures, clearly indicate the superiority of *Sinusoidal* initialization. Specifically, this approach consistently achieved higher final validation accuracy and accelerated convergence rates compared to SOTA stochastic initialization methods. Our results quantify an average accuracy improvement over the default initialization of approximately $4.9\%$ and a notable boost in convergence speed of around $20.9\%$, underscoring the method's robustness and versatility.

In conclusion, our findings challenge the longstanding assumption that randomness is essential for effective model initialization, presenting strong theoretical and empirical arguments in favor of deterministic methods. The *Sinusoidal* initialization represents a significant breakthrough in DNN initialization practices, paving the way for future explorations into deterministic initialization schemes, potentially redefining standard practices in the design and training of DNNs.

## Acknowledgments and Disclosure of Funding

This research was funded by the projects PID2023-146569NB-C21 and PID2023-146569NB-C22 supported by MICIU/AEI/10.13039/501100011033 and ERDF/UE. Alberto Fernández-Hernández was supported by the predoctoral grant PREP2023-001826 supported by MICIU/AEI/10.13039/501100011033 and ESF+. Jose I. Mestre was supported by the predoctoral grant ACIF/2021/281 of the Generalitat Valenciana. Manuel F. Dolz was supported by the Plan Gen–T grant CIDEXG/2022/013 of the Generalitat Valenciana.

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

# A  Limitations and Societal Impact

This work opens the door to exploring deterministic initializations that exploit symmetry to endow neural networks with desirable properties beyond the reach of stochastic schemes. Among the possible designs, we focused on the *Sinusoidal* initialization due to its natural structure and analytical tractability. Nevertheless, the broader space of structured deterministic initializations remains scarcely explored, offering a fertile avenue for future research.

In terms of empirical evaluation, while our method consistently outperforms alternatives across most settings, results on MobileNetV3 fall slightly behind those of Orthogonal initialization. Understanding why this architecture deviates from the general trend, despite its apparent similarity to EfficientNetV2, could reveal important insights and guide further refinements. We regard this as a localized irregularity rather than a fundamental shortcoming.

Finally, although we have a clear theoretical understanding of how *Sinusoidal* initialization guarantees the absence of *skewed* neurons in MLPs regardless of depth, and unlike traditional stochastic methods, further research is needed to fully characterize this behavior across a broader range of architectures. In particular, establishing a general theoretical framework that ensures the absence of *skewed* neurons in all established architectures remains an open and promising direction for future work.

Regarding the societal impact, the proposed *Sinusoidal* initialization enables faster convergence during training, reducing the number of epochs required to reach target performance. This directly translates into lower computational demands, which in turn reduces both the economic cost and the environmental footprint of training DNNs.

Given that a substantial portion of the energy consumed in large-scale machine learning pipelines originates from high-performance computing centers, many of which still rely on carbon-emitting energy sources, the adoption of more efficient initialization strategies could contribute to a reduction in overall $CO_2$ emissions. While the exact magnitude of this impact depends on deployment scale and infrastructure specifics, our results highlight a concrete opportunity to align algorithmic efficiency with broader sustainability goals.

# B  Theoretical details

This appendix provides the theoretical foundations underlying the results stated in Section 3, and in particular formal proofs of Theorems 1, 2 and 3. We also present a general version of Theorem 1, extending the simplified setting used in the main text to a more flexible framework that covers non-identically distributed inputs and milder moment assumptions.

Beyond these formal results, we include further technical insights that clarify the connection between random initializations and the emergent activation patterns observed in the early layers of deep neural networks.

## B.1  Threshold equivalence

In Theorem 1 we stated, under strong i.i.d. assumptions, that the weighted sum $Z_n = \sum W_i X_i$ is controlled with high precision by the magnitude of the total weight $S_n = \sum W_i$. Specifically, for each level $\alpha \in (0, 1/2)$, there exists a sharp threshold $\lambda_n(\alpha)$ such that the probability $P(Z_n > 0 \mid W_1, \ldots, W_n)$ deviates from $1/2$ by more than $\alpha$ if and only if $|S_n| > \lambda_n(\alpha)$, with high probability.

In this subsection, we show that this result admits a broader formulation under significantly weaker assumptions. In particular, the data $X_1, \ldots, X_n$ need not be identically distributed, and only mild regularity is required. The result below extends Theorem 1, and justifies its asymptotic equivalence through general concentration and normalization arguments.

**Theorem 4** (Threshold Equivalence, general version)**.** *Let* $(W_1, X_1), \ldots, (W_n, X_n)$ *be independent pairs of real random variables, where the weights* $W_1, \ldots, W_n$ *are i.i.d., independent of the data* $X_1, \ldots, X_n$, *and satisfy*

$$\mathbb{E}[W_1] = 0, \quad \text{Var}(W_1) = \theta^2 \in (0, \infty).$$

*Assume that the data $X_1, \ldots, X_n$ are independent with a common mean $\mu \neq 0$, variances $\mathrm{Var}(X_i) = \sigma_i^2 \in (0, \infty)$, and that*

$$\frac{1}{n} \sum_{i=1}^{n} \sigma_i^2 \longrightarrow \bar{\sigma}^2 \in (0, \infty), \quad \text{as } n \to \infty.$$

*Finally, assume the conditional Lindeberg condition,*

$$\frac{1}{\sum_{i=1}^{n} W_i^2 \sigma_i^2} \sum_{i=1}^{n} \mathbb{E}\left[ W_i^2 (X_i - \mu)^2 \mathbf{1}_{\{|X_i - \mu| > \varepsilon \sqrt{\sum W_i^2 \sigma_i^2}\}} \right] \longrightarrow 0 \quad \text{for all } \varepsilon > 0,$$

*holds almost surely. Define $S_n = \sum_{i=1}^{n} W_i$ and $Z_n = \sum_{i=1}^{n} W_i X_i$. Then for any $\alpha \in (0, 1/2)$, define*

$$\lambda_n(\alpha) := \left( \theta \sqrt{\bar{\sigma}^2} \sqrt{n} \right) / \mu \cdot \Phi^{-1} \left( 1/2 + \alpha \right),$$

*where $\Phi^{-1}$ is the quantile function of the standard normal distribution. Then, as $n \to \infty$, the following equivalence holds with probability tending to one:*

$$|P(Z_n > 0 \mid W_1, \ldots, W_n) - 1/2| > \alpha \quad \text{if and only if} \quad |S_n| > \lambda_n(\alpha).$$

In particular, for every $\alpha \in (0, 1/2)$, the threshold $\lambda_n(\alpha)$ is asymptotically unique with high probability.

**Sufficient conditions.** The result holds in particular when the data $X_1, \ldots, X_n$ are i.i.d. with $\mathbb{E}[X_1] = \mu \neq 0$ and $\mathrm{Var}(X_1) = \sigma^2 < \infty$. In this case, the conditional Lindeberg condition is automatically satisfied by the classical Lindeberg–Feller Central Limit Theorem (CLT), since the summands $W_i X_i$ form a triangular array with independent rows and uniformly bounded second moments. These are precisely the assumptions stated in Theorem 1, which is therefore recovered as a particular case of the present general result.

*Proof.* Let us write $\Sigma_n^2 := \sum_{i=1}^{n} W_i^2 \sigma_i^2$. Given the weights $W_1, \ldots, W_n$, we consider the normalized sum

$$(Z_n - \mu S_n)/\Sigma_n \mid W_1, \ldots, W_n.$$

Under the assumptions of the theorem, the classical Lindeberg–Feller CLT for triangular arrays implies that this converges in distribution to a standard normal:

$$(Z_n - \mu S_n)/\Sigma_n \mid W_1, \ldots, W_n \xrightarrow{\mathcal{D}} \mathcal{N}(0, 1).$$

As a consequence,

$$P(Z_n > 0 \mid W_1, \ldots, W_n) = \Phi\left( \mu S_n / \Sigma_n \right) + o(1),$$

where the error $o(1)$ vanishes in probability as $n \to \infty$.

Now observe that since $W_1, \ldots, W_n$ are i.i.d. with variance $\theta^2$, we have by the law of large numbers that

$$\sum_{i=1}^{n} W_i^2 = n\theta^2 (1 + o(1)), \quad \text{and} \quad \Sigma_n^2 = \sum_{i=1}^{n} W_i^2 \sigma_i^2 = n\theta^2 \bar{\sigma}^2 (1 + o(1)).$$

Combining these gives

$$\frac{\mu}{\Sigma_n} = \frac{\mu}{\theta \sqrt{\bar{\sigma}^2}} \cdot \frac{1}{\sqrt{n}} (1 + o(1)) := c_n,$$

so that

$$P(Z_n > 0 \mid W_1, \ldots, W_n) = \Phi(c_n S_n) + o(1).$$

Now fix any $\alpha \in (0, 1/2)$. Since the standard normal cdf $\Phi$ is strictly increasing and symmetric about zero, we find that

$$|P(Z_n > 0 \mid W_1, \ldots, W_n) - 1/2| > \alpha \quad \text{if and only if} \quad |c_n S_n| > \Phi^{-1}\left(1/2 + \alpha\right).$$

Solving for $|S_n|$, this is equivalent to

$$|S_n| > \Phi^{-1}\left(1/2 + \alpha\right) / c_n = \lambda_n(\alpha)(1 + o(1)).$$

This equivalence holds with probability tending to one, since the approximation error vanishes and $S_n$ is independent of the remainder.

Finally, note that the monotonicity of $\Phi$ implies that this inequality can hold for at most one threshold $\lambda_n(\alpha)$. Therefore, the threshold is asymptotically unique. $\square$

**Theorem** ([Theorem 2 (Asymptotic skewness under random initialization)). *] Consider a neuron with zero bias and input $X_1, \ldots, X_n$ and weights $W_1, \ldots, W_n$. Assume $\{W_i\}$ are i.i.d. with $\mathbb{E}[W_1] = 0$ and $\mathrm{Var}(W_1) = \theta^2 \in (0, \infty)$, and $\{X_i\}$ are i.i.d., independent of $\{W_i\}$, with $\mathbb{E}[X_1] = \mu > 0$ and $\mathrm{Var}(X_1) = \sigma^2 < \infty$. Let $Z_n = \sum_{i=1}^{n} W_i X_i$. Then for every deterministic sequence $\alpha_n \downarrow 0$, the probability of having*

$$\left| P(Z_n > 0 \mid W_1, \ldots, W_n) - 1/2 \right| > \alpha_n$$

*tends to one as $n \to \infty$. Equivalently, with probability tending to one there exists a nonzero skewness level at which the neuron is skewed.*

*Proof.* Write $S_n = \sum_{i=1}^{n} W_i$ and fix any sequence $\alpha_n \downarrow 0$. By the Threshold Equivalence Theorem 1, for each fixed $\alpha \in (0, 1/2)$ the event

$$E_n(\alpha) := \left\{ \left| \mathbb{P}(Z_n > 0 \mid W_1, \ldots, W_n) - 1/2 \right| > \alpha \right\}$$

is, with probability tending to one, equivalent to

$$F_n(\alpha) := \left\{ |S_n| > \lambda_n(\alpha) \right\},$$

where

$$\lambda_n(\alpha) = (\theta \sigma \sqrt{n})/\mu \cdot \Phi^{-1}(1/2 + \alpha).$$

To apply this along a vanishing sequence, choose a decreasing sequence $\alpha^{(k)} \downarrow 0$ and integers $N_k \uparrow \infty$ such that, for all $n \geq N_k$, the difference between $E_n(\alpha^{(k)})$ and $F_n(\alpha^{(k)})$ has probability at most $2^{-k}$. Define a piecewise constant selection $\alpha_n = \alpha^{(k)}$ for $N_k \leq n < N_{k+1}$. This ensures

$$\mathbb{P}\big( E_n(\alpha_n) \triangle F_n(\alpha_n) \big) \longrightarrow 0.$$

It remains to show that $\mathbb{P}\big( F_n(\alpha_n) \big) \to 1$. By the Central Limit Theorem,

$$\frac{S_n}{\theta \sqrt{n}} \Rightarrow \mathcal{N}(0, 1).$$

Since $\alpha_n \to 0$ and $\Phi^{-1}$ is continuous at $1/2$, we have that

$$c_n := \frac{\lambda_n(\alpha_n)}{\theta \sqrt{n}} = \frac{\sigma}{\mu} \, \Phi^{-1}\big( \tfrac{1}{2} + \alpha_n \big) \longrightarrow 0.$$

Thus, if $\mathcal{N}(0, 1)$ is denoted by $G$, it follows that

$$\mathbb{P}\big( F_n(\alpha_n) \big) = \mathbb{P}\left( \left| \frac{S_n}{\theta \sqrt{n}} \right| > c_n \right) \longrightarrow \mathbb{P}(|G| > 0) = 1.$$

Combining both steps yields

$$\mathbb{P}\big( E_n(\alpha_n) \big) \geq \mathbb{P}\big( F_n(\alpha_n) \big) - \mathbb{P}\big( E_n(\alpha_n) \triangle F_n(\alpha_n) \big) \longrightarrow 1,$$

and the result follows. $\qquad\qquad\square$

**Propagation of *skewness* across layers.** At the input layer, the assumption of equal means across coordinates, central to Theorem 4, is typically satisfied due to standard data normalization practices. However, when standard stochastic initializations are used, the first hidden layer introduces an unavoidable imbalance: the randomness in the initialization causes the activations to deviate from a centered distribution, resulting in the emergence of *skewed* neurons. Consequently, the inputs to the second layer are no longer mean-centered across dimensions. As training progresses deeper into the DNN, these initial asymmetries are compounded, each layer propagates and potentially amplifies the imbalances introduced by the previous one.

This effect gradually breaks the correspondence between the *skewness* of a neuron and the statistic $|S|$, as established in Theorem 4. In particular, the conditional expectation $\mathbb{E}[Z_n \mid W_1, \ldots, W_n] = \mu S_n$ hinges critically on the input means $\mu_i$ being equal. Once this condition fails, the quantity $\sum_i \mu_i W_i$ governing the conditional bias of the neuron is no longer proportional to $S_n$, and the connection between sign bias and the statistic $S$ deteriorates.

This breakdown is clearly observable in the evolution of the histogram shown in Figure 7, which displays the distribution of the *skewness* statistic $S$ across layers in a multilayer perceptron with four hidden layers. If a layer is initiated with a stochastic criterion (such as Glorot), the distribution of $S$ closely follows a normal distribution, as predicted in Section 3. At the first hidden layer initiated with a random distribution (first row and first column of Figure 7), *skewed* neurons clearly remain concentrated in the tails. However, as depth increases (second, third and fourth histograms in the first row), these blue tails representing *skewed* neurons progressively dissolve and spread toward the center of the distribution. As a result, *skewed* neurons begin to appear even at small values of $S$, indicating that neuron *skewness* becomes less predictable from the weight imbalance alone.

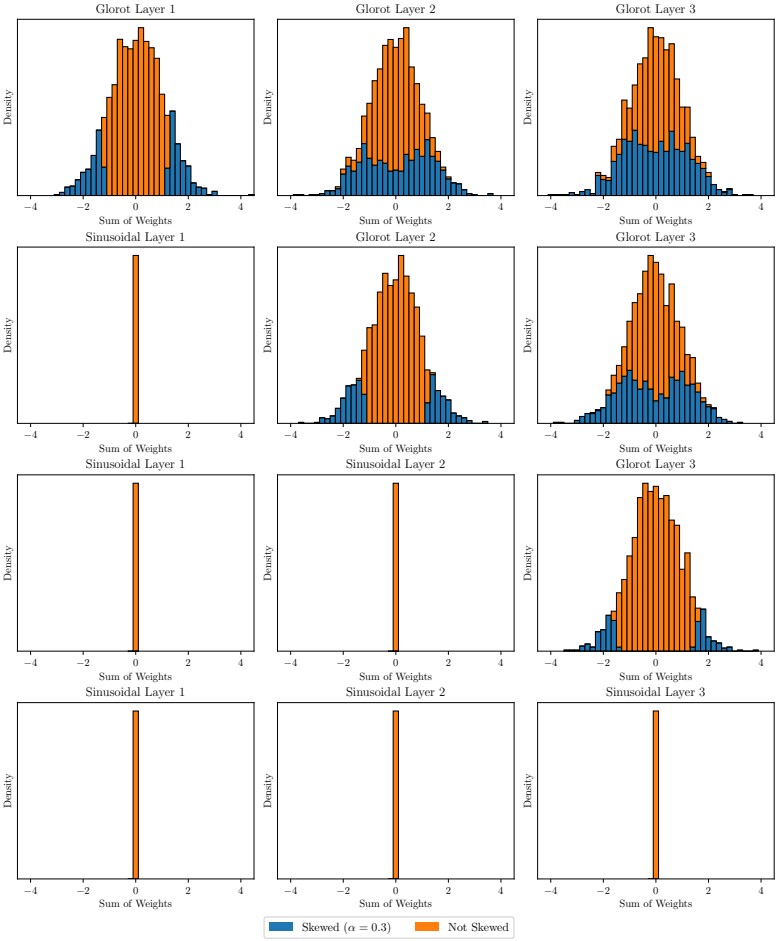

Figure 7: Evolution of the values of the statistic $S$ for *skewed* neurons across layers in a MLP.

By contrast, the *Sinusoidal* initialization fundamentally avoids this pathology. As shown in Theorem 3, each row of the initialization matrix sums to zero, ensuring that the output of the first layer is strictly centered around zero for all neurons (first element of second row). This structural symmetry guarantees that no directional bias is introduced at the very start of the network. More importantly, it prevents the cascade of asymmetries that otherwise accumulates across layers. The rest of the figure shows how *Sinusoidal* initializations in the first layers of the model help in progressively achieving an equilibrium in terms of *skewed* neurons. As a result, the relationship between *skewness* and $S$ remains intact even in deeper layers, and the network preserves maximal discriminatory power throughout its depth. This stability is reflected both theoretically and empirically: in Table 1, the proportion of *skewed* neurons under *Sinusoidal* initialization remains identically zero across all tested thresholds. This key structural ingredient, the enforced symmetry at initialization, lays the foundation for a qualitatively new regime in network design. It resolves long-standing inefficiencies related to neuron utilization and enables networks to start from a maximally expressive configuration, thereby accelerating training dynamics systematically. As such, the *Sinusoidal* initialization marks a

turning point in the initialization literature, offering a principled and effective alternative to stochastic schemes.

## B.2 Row cancellation in the Sinusoidal initialization

As introduced in Section 3, the *Sinusoidal* initialization induces a deterministic structure on the weight matrix that stands in stark contrast with the randomness of classical schemes. This structure is not only symmetric, but also highly regular in a way that yields exact cancellation patterns across the rows of the initialization. In particular, each neuron's incoming weights sum to zero, leading to a strictly balanced preactivation regime from the very first forward pass.

The result we now prove formalizes this cancellation property. To that end, let us recall how the weights are defined under the *Sinusoidal* initialization. Let $W \in \mathbb{R}^{m \times n}$ be defined coordinate-wise as

$$W[i,j] = a \cdot \sin(k_i \cdot x_j + \phi_i),$$

where

$$k_i = 2\pi i, \ x_j = j/n \ \text{ and } \ \phi_i = 2\pi i/m,$$

for $i \in \{1, \ldots, m\}$ and $j \in \{1, \ldots, n\}$.

**Theorem** (Theorem 3 (Row Cancellation)). *Let* $W \in \mathbb{R}^{m \times n}$, *with* $m < n$, *be defined as in Equation* (1). *Then, for every row index* $i \in \{1, \ldots, m\}$, *the entries along that row sum to zero:*

$$\sum_{j=1}^{n} W[i,j] = 0.$$

*Proof.* Fix any $i \in \{1, \ldots, m\}$. We consider the sum:

$$\sum_{j=1}^{n} W[i,j] = a \sum_{j=1}^{n} \sin\left(2\pi i \cdot \frac{j}{n} + \frac{2\pi i}{m}\right).$$

Applying the identity $\sin(\alpha + \beta) = \sin\alpha\cos\beta + \cos\alpha\sin\beta$, we rewrite the sum as

$$\sum_{j=1}^{n} \sin\left(2\pi i \cdot \frac{j}{n} + \frac{2\pi i}{m}\right) = \cos\left(\frac{2\pi i}{m}\right)\sum_{j=1}^{n}\sin\left(\frac{2\pi i j}{n}\right) + \sin\left(\frac{2\pi i}{m}\right)\sum_{j=1}^{n}\cos\left(\frac{2\pi i j}{n}\right).$$

Thus, to conclude the proof, it suffices to show that both trigonometric sums vanish, that is,

$$\sum_{j=1}^{n} \sin\left(\frac{2\pi i j}{n}\right) = 0, \quad \sum_{j=1}^{n} \cos\left(\frac{2\pi i j}{n}\right) = 0.$$

To show this, let $\omega \in \mathbb{C}$ be the unitary complex number and angle $2\pi i/n$, which clearly satisfies that $w^n = 1$. As $i \leq m < n$, it follows that $\omega \neq 1$, and hence

$$\sum_{j=1}^{n} w^j = \omega \frac{1 - \omega^n}{1 - \omega} = 0.$$

The two real-valued sums above correspond to the imaginary and real parts of this complex sum, respectively. Therefore, both must vanish, and the result follows. $\qquad\square$

## C  Experimental Setup

This appendix contains additional material complementing the main text. Specifically, it provides complete details for all experiments presented in the paper, including training configurations and hardware specifications.

For full reproducibility, all code used to conduct the experiments is publicly available in the accompanying GitHub repository: `https://github.com/`.

To assess the robustness of our findings, all experiments supporting the main claims were repeated three times. We report mean values across these runs. Standard deviations were computed and found to be consistently small. As these deviations do not materially affect the conclusions, they are omitted from the main tables and figures for clarity.

## C.1 Models, datasets and initializations

This work employs a range of DNN architectures across vision and language tasks. The models and datasets used are listed below, along with their sources and brief descriptions.

**Models**

- **ResNet-50** (Torchvision): A deep residual network with 50 layers, widely used for image classification tasks.
- **MobileNetV3 small** (Torchvision): A lightweight convolutional neural network optimized for mobile and low-resource environments.
- **EfficientNetV2 small** (Torchvision): A family of convolutional models optimized for both accuracy and efficiency.
- **ViT B16** (Torchvision): A Vision Transformer model that applies transformer architectures to image patches.
- **BERT-mini** (Hugging Face): A compact version of BERT for natural language processing tasks, trained with the Masked Language Modeling (MLM) objective.

**Datasets**

- **CIFAR-100** (Torchvision): A dataset of 60,000 32×32 color images in 100 classes, with 600 images per class.
- **Tiny-ImageNet** (Kaggle): A subset of ImageNet with 200 classes and 64×64 resolution images, sourced from the Kaggle repository `xiataokang/tinyimagenettorch`.
- **ImageNet-1k** (Official Repository): A large-scale dataset containing over 1.2 million images across 1,000 categories, used for benchmarking image classification models.
- **WikiText2** (HuggingFace Datasets): A small-scale dataset for language modeling, using the `wikitext-2-raw-v1` version to preserve raw formatting.

**Initialization Procedures**    Four initialization schemes were considered in this work:

- **Default**: This corresponds to the initialization procedure provided in the original codebase for each model, as implemented in the respective libraries (e.g., Torchvision or Hugging Face).
- **Orthogonal** and **Sinusoidal**: These initializations are manually applied to the weights of all convolutional and linear layers in the model.
- **LSUV (Layer-Sequential Unit Variance)**: Implemented using the authors' original repository (github.com/ducha-aiki/lsuv). This method is applied to convolutional and linear layers and requires a single batch of data from the associated dataset to initialize.

All resources were used in accordance with their respective licenses and usage policies. Note that all dataset splits used correspond to the default configurations

## C.2 Activation Imbalance Evaluation

This subsection provides a detailed description of the experiments underlying the theoretical insights discussed in Section 3, particularly those supporting Table 1 and Figures 4 and 5.

**Experiment 1: Neuron Activation States (ViT)**

To analyze the activation state of neurons, we used the Vision Transformer (ViT-B/16) model trained on ImageNet-1k. The model was initialized using each of the initialization schemes considered in this work, and a forward pass was performed using 768 randomly sampled images from the ImageNet-1k validation set. We captured the output of the second linear layer in the feed-forward block of the last encoder layer—i.e., the final layer before the classifier—resulting in a tensor of shape $768 \times 768$ (samples × neurons). To enable visual representation within the paper, we subsampled this tensor to

250 neurons and 250 samples, preserving the structure and interpretability of the activation patterns. This experiment provides the data used in Figure 4.

**Experiment 2: Statistic $S$ and *skewness* Relationship and OUI**

To isolate the relation between the statistic $S$ and neuron *skewness*, we constructed a synthetic feedforward model consisting of a 3-layer MLP with the structure: ReLU → Linear → ReLU → Linear → ReLU → Linear. The input data consisted of random vectors sampled from a Normal distribution. This controlled setting enables clear observation of activation statistics before depth-related effects dilute the signal, as discussed in Appendix B.1.

The histogram in Figure 5 was generated using the outputs from the first linear layer, where the relationship with the statistic $S$ is still prominent. In contrast, the statistics reported in Table 1—such as the percentage of *skewned* neurons at significance levels $\alpha = 0.1$ and $0.3$, and the OUI was computed from the final linear layer in the MLP, where cumulative effects of initialization are more pronounced.

We confirmed the robustness of these results by also sampling inputs from a Uniform distribution, observing consistent trends in *skewness* behavior and $S$-statistic correlation.

### C.3 Benchmark Evaluation

This subsection provides the full experimental details for the training process corresponding to the experiments presented in Section 4, and summarized in Table 3 and Figures 1, 6, and 11. The tables below detail the training hyperparameters (Table 4) and hardware (Table 5) used for each model-dataset pair.

Table 4: Training hyperparameters for all evaluated models.

| Model | Dataset | Epochs | Optimizers | LR | WD | Batch size |
|---|---|---|---|---|---|---|
| ResNet-50 | CIFAR-100 | 100 | SGD, Adam, AdamW | $10^{-3}$ | $10^{-3}$ | 64 |
| MobileNetV3 | Tiny-ImageNet | 200 | SGD, Adam, AdamW | $10^{-3}$ | $10^{-3}$ | 64 |
| EfficientNetV2 | Tiny-ImageNet | 200 | SGD, Adam, AdamW | $10^{-3}$ | $10^{-3}$ | 64 |
| ViT-B16 | ImageNet-1k | 100 | SGD | $10^{-3}$ | $10^{-3}$ | 64 |
| BERT-mini | WikiText | 200 | AdamW | $5 \cdot 10^{-5}$ | $10^{-3}$ | 16 |

Table 5: Hardware and runtime per single training experiment.

| Model | Dataset | GPU | Training time (hours) |
|---|---|---|---|
| ResNet-50 | CIFAR-100 | NVIDIA A100-SXM4-80GB | 2 |
| MobileNetV3 | Tiny-ImageNet | NVIDIA A100-SXM4-80GB | 4 |
| EfficientNetV2 | Tiny-ImageNet | NVIDIA A100-SXM4-80GB | 4 |
| ViT-B16 | ImageNet-1k | NVIDIA H100-PCIe-94GB | 168 |
| BERT-mini | WikiText | NVIDIA A100-SXM4-80GB | 2 |

## D  Additional experimental results

### D.1  Arcsine random initialization

To further validate that the key factor behind the effectiveness of our *Sinusoidal* initialization is the structure it imposes, rather than its unusual distribution, we performed an additional experiment using a randomized initialization. In this variant, weights were sampled independently from the arcsine distribution, which matches the distribution of our *Sinusoidal* initialization but lacks its balanced structure.

We replicated the experiments from Figure 4 and Figure 5 using this randomized initialization. The results, shown in Figure 8, demonstrate that this initialization behaves similarly to other stochastic methods: the activations exhibit comparable neuron *skewness*. This confirms that the stochastic nature of the weights, not the distribution itself, is responsible for the observed effects.

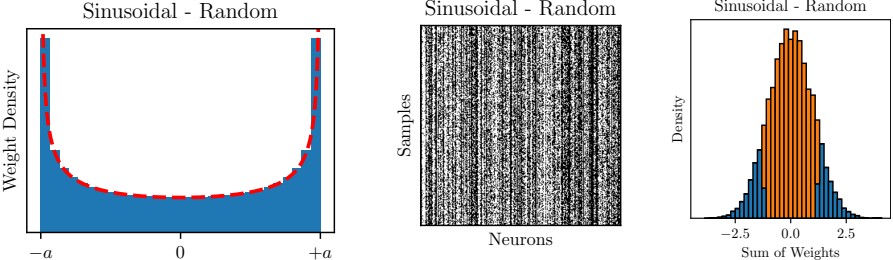

Figure 8: Weight distribution, neuron activation states and histogram comparing the statistic S values with neuron *skewness* at $\alpha = 0.3$ for a random initialization with arcsine distribution.

These findings make it clear that the structure introduced by the *Sinusoidal* initialization is the critical component driving its benefits. This outcome is consistent with the theory established in Subsection 3.2, and further reinforces that the deterministic layout of weights plays a central role in mitigating activation *skewness*. To further support this conclusion, we replicated the experiment reported in Section 4 on ResNet-50 with CIFAR-100 and SGD. The training results, presented in Figure 9, clearly show how the randomness of the arcsine variant delays convergence with respect to the structured *Sinusoidal* initialization, which provides a faster training process.

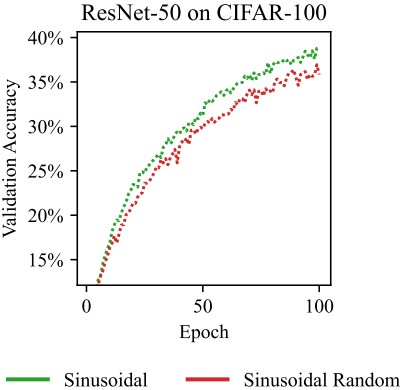

Figure 9: Training curves of the *Sinusoidal* initialization and its randomized counterpart, showing that the structural arrangement of weights accelerates the initial stages of training.

## D.2 Comparison with other deterministic initializations

In addition to stochastic baselines, a few deterministic initialization schemes have also been proposed in the literature, namely those introduced in Zhao et al. [2022] and Blumenfeld et al. [2020], already referenced in Section 2. To illustrate the relative performance of these methods, we provide a representative experiment on ResNet-50 with CIFAR-100 using SGD, under the same hyperparameter configuration described in Section 4. While this comparison is restricted to a single training setup, it provides a direct assessment of how these deterministic strategies behave under identical conditions. The results are shown in Figure 10.

As can be observed, the *Sinusoidal* initialization clearly outperforms both alternatives, achieving faster convergence and higher final accuracy. This experiment thus provides compelling evidence that, within the family of deterministic approaches, our method offers a distinct advantage.

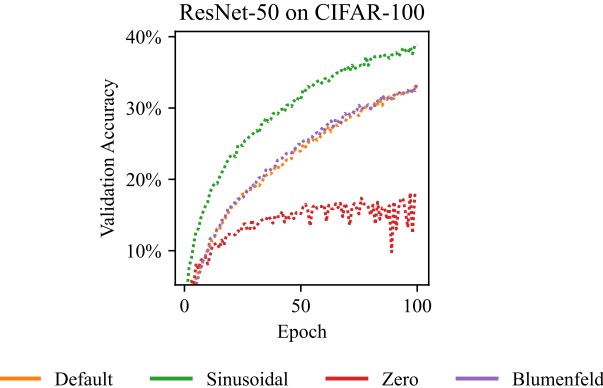

Figure 10: Training curves of *Default*, *Sinusoidal*, *Zero* and *Blumenfeld* initializations, showing that our approach consistently outperforms the other deterministic methods.

## D.3 Additional Training Plots

To complement the main experimental results presented in Section 4, we include in Figure 11 a detailed comparison of the training dynamics between the default and *Sinusoidal* initialization schemes. For each model and dataset combination, and for every optimizer used during training, we plot both the loss and accuracy curves across epochs.

This visualization allows for a direct assessment of convergence behavior and generalization performance throughout training. Notably, the figure highlights differences in early-stage convergence speed, final validation accuracy levels, and training loss smoothness under the two initialization schemes.

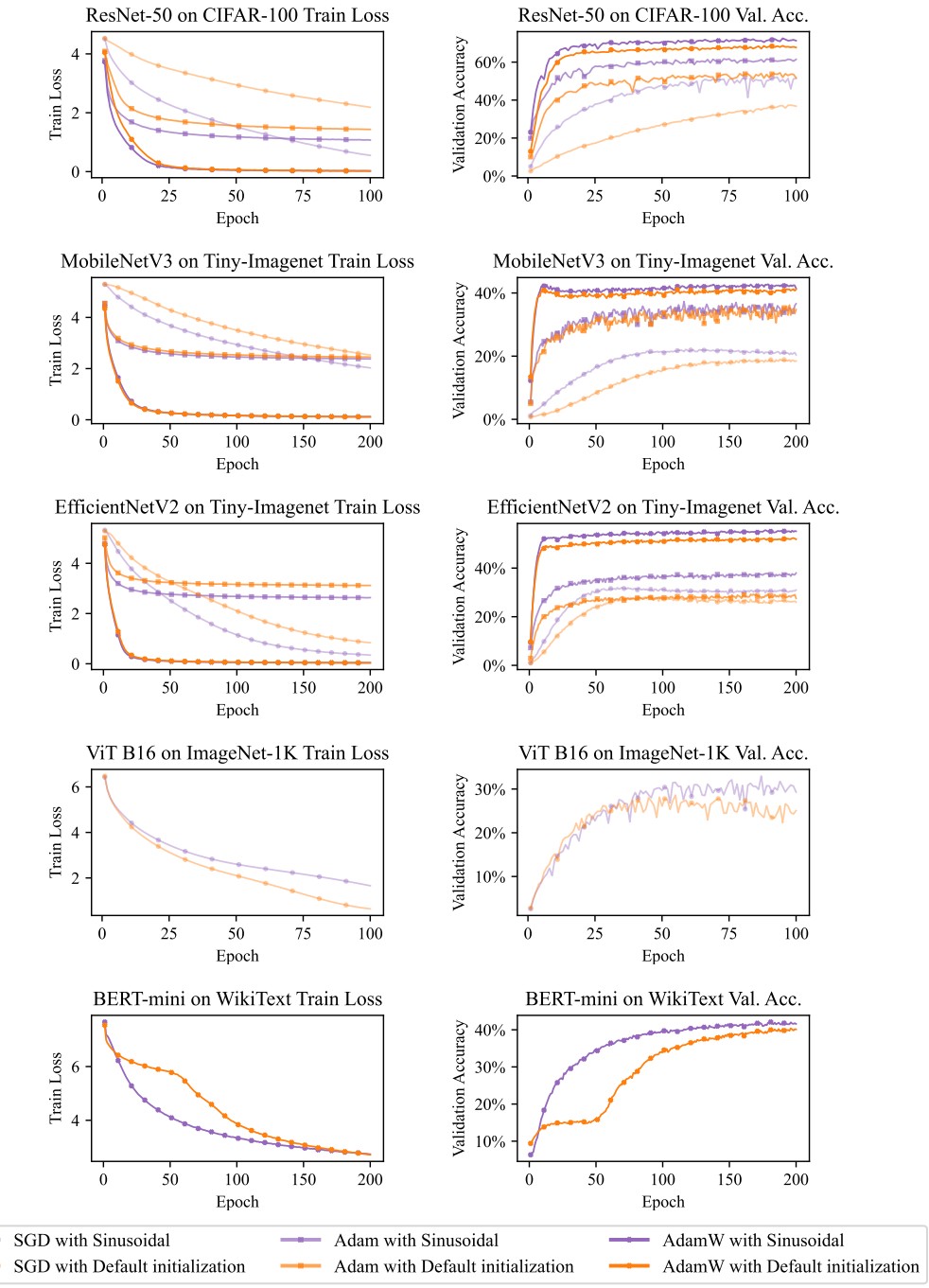

Figure 11: Training loss and accuracy curves for all model-dataset pairs using default and *Sinusoidal* initialization. Each subplot shows the result for a particular model-optimizer combination.

