# OpenReview forum: "Sinusoidal Initialization, Time for a New Start"
_NeurIPS.cc/2025/Conference — NeurIPS 2025 poster_

### Official Review · Reviewer_kR57 · 2025-06-12

**Clarity:** 3
**Significance:** 2
**Originality:** 3
**Rating:** 5
**Confidence:** 4

**Summary:**

In this paper the authors propose a deterministic initialization scheme for neural networks that removes randomness and improves the convergence and final accuracy characteristics by lowering the number of "skewed" neurons. Authors provide both theoretical justifications as well as some experimental results to support their claims.

**Questions:**

-  Authors claim that randomness is a negative aspect in initialization. Could the authors provide stronger evidence or justification for this claim? How do you account for the possibility that stochastic initializations achieving the same target distribution (e.g., arcsine) might perform similarly?

- How do the authors explain the substantial drop in performance for MobileNetV3 with SGD (28% to 21%)? Are there specific conditions where your method is expected to degrade performance?

- Why was ImageNet-1k only evaluated with a ViT transformer?

- Can the authors clarify under what circumstances or configurations you expect your proposed initialization method to improve performance? Are there cases where it should not be applied?

- How does the impact of  the proposed initialization change with smaller architectures? Have the authors conducted any experiments on such architectures?

- In Table 3, authors report "Maximum accuracy" but label it as "accuracy at convergence." Could the authors clarify this discrepancy?

**Ethical Concerns:**

["NO or VERY MINOR ethics concerns only"]

**Final Justification:**

Overall, I find this paper inspiring and would like to see it accepted. Authors clarrified one important concern that I had and I am increasing my rating in response. The validation could be stronger, as the other reviewers noted.

**Limitations:**

Yes.

**Paper Formatting Concerns:**

No.

**Quality:**

3

**Strengths And Weaknesses:**

First, I would like to say that I really enjoyed reading this paper. In the saturated landscape of minor changes over existing approaches, the paper provides a new perspective into a less studied and often overlooked aspect of deep neural networks that could have a significant impact on the performance of the models. The paper is clearly written and easy to follow. Authors present all concepts in a clear and very understandable way and provide appropriate motivation for their work.

However, despite really liking the paper and the idea, I am afraid that the paper is significantly lacking in the validation part. Authors only evaluated five different combinations of architectures/datasets. Even for these selected combinations the demonstrated performance improvements do not always match the authors’ claims, apart from the initial convergence acceleration. More specifically, I have the following comments:

- Authors claim at various points that randomness is actually a negative aspect in the initialization of networks. I am quite skeptical of this claim and I would like to invite authors to revisit this claim. This work demonstrates that a more careful initialization scheme can reduce the number of skewed neurons, leading to better performance. The method happens to be deterministic, yet this does not prove that stochastic initialization cannot achieve the same results. For example, it might be reasonable to assume that any initialization that leads to the same arcsine distribution (as shown in Fig.2) would probably perform the same, unless authors provide evidence against this.

- ImageNet-1k is only evaluated in a ViT transformer. I think evaluating a ResNet architecture on this dataset is essential and shouldn't be more computationally complex. This is a major concern.

- In a few cases, the performance is substantially lower. For example, for a MobileNetV3 using SGD, the max accuracy is reduced from 28% to 21%. This result, along with the lack of extensive experiments, creates some questions on whether the authors' claims are universally true or are valid only in some cases. Authors should better indicate when they expect improvements by applying the proposed initialization approach. It might be useful to connect the performance improvements to the designed metrics (e.g., skewed neurons).

- Furthermore, not all networks seem to have converged, with Adam/AdamW-trained networks showing substantially better performance. I understand that authors, in their desire to have a fair setup, might have reduced the amount of hyper-parameter tuning performed. However, the vast differences between the employed configurations for the same architecture/dataset show that some setups might be far from optimal and some networks might not have yet converged. Therefore, some of the results might actually still show the performance at more initial stages of the convergence instead of the final one.

- The paper lacks evaluation experiments on smaller architectures. Does the impact of initialization diminish as the architecture gets smaller? This is a minor issue.

- In Table 3 authors claim that they report accuracy "at convergence," yet "Maximum accuracy" is reported, which is not the same as the accuracy at convergence, if I understand the concepts correctly. This is a minor issue.

Even though it seems that the results provided in the paper might be a bit premature, I would tend towards acceptance, unless there are also important concerns raised by the rest of the reviewers.

---

> ### Author Rebuttal · Authors · 2025-07-29
>
> We sincerely thank the reviewer for the extremely thoughtful comments, both the positive ones and the critical ones. We truly appreciate the encouraging words about the **clarity** and **motivation** of the paper, and we are equally grateful for the **constructive feedback**, which will undoubtedly help us improve the work further.
>
> ----
>
> ## **Question 1: On the role of randomness in initialization**
>
> We completely agree with this comment. In fact, we included an appendix discussing a stochastic initialization that also follows the arcsine distribution, similarly to our deterministic proposal, and we showed theoretically that this variant still results in a non-negligible proportion of skewed neurons. While we have not yet implemented or tested this variant empirically, we plan to conduct **small- and medium-scale experiments** to validate this behavior in practice and to **report the results in the corrected version** in the same appendix where this topic is addressed. We hope this will help clarify, both theoretically and experimentally, that the key factor is not the distribution itself, but rather the **disorder induced by the stochastic process**, which leads to the emergence of skewed neurons.
>
> ----
>
> ## **Question 2: On evaluating ResNet on ImageNet-1K**
>
> Agreed, and we appreciate the suggestion. While we fully recognize the value of including a ResNet experiment on ImageNet-1K, we also note that space constraints and the need to balance coverage across different architectures led us to focus on a **selected set of pairs of models and datasets**. Our goal was to include representative scenarios rather than exhaustively testing all combinations. That said, we consider this an excellent **addition for future work** or extended versions of the manuscript.
>
> ----
>
> ## **Question 3: On the drop in performance for MobileNetV3 with SGD**
>
> This is a fair point. We have identified that the main issue here is the **Swish activation** used in MobileNetV3. We repeated the same experiments using *ReLU* instead, and observed **clear superiority of Sinusoidal** against the other initialization methods. We decided to keep the original experiment in the results section to remain fully transparent; while we could have omitted it, we believe doing so would have missed the point. This case highlights an interesting interaction that deserves **further investigation**. In general, our theoretical insights suggest that controlling the proportion of skewed neurons leads to **improved convergence speed and final accuracy** by initializing the network in a more favorable regime. However, this interaction is still under investigation, especially in architectures where the non-linearities or structural elements deviate from the *linear \+ activation* pattern. We are **including in the manuscript an annotation** discussing the results obtained with MobileNet using *ReLU*, along with a note addressing the considerations mentioned above. Despite this, we believe the strong results on the other architectures and tasks are sufficient to consider this a **mature and promising contribution to the literature**.
>
> ----
>
> ## **Question 4: On possible lack of convergence in some experiments**
>
> In all our setups, to fairly compare initializations using AUC across different runs, we ensure that all models are trained **for the same number of epochs**. For each combination of model, dataset, and optimizer, training is performed with early stopping to **guarantee convergence** for every initialization. Once the slowest run has converged, the remaining models continue training for the same number of additional epochs until reaching the maximum epoch count. This procedure ensures that (1) all trainings **reach their maximum validation accuracy**, and (2) the number of epochs used to calculate the AUC for measuring convergence speed **is consistent** across initializations. The differences in final performance across optimizers are expected: some optimizers simply reach better minima than others. However, our intention is to compare initializations for fixed optimizers, rather than comparing optimizers per se. We are adding a **clarification to the paper** describing this methodology to prevent any possible misunderstanding about whether the experiments have properly converged.
>
> ----
>
> ## **Question 5: On the absence of small-scale architectures**
>
> We appreciate this comment and are going to add an experiment using *LeNet-5* on CIFAR-10 to confirm that our initialization remains beneficial at a small scale. Although the absolute gains in time or accuracy are smaller in such settings, this experiment supports the **down-scaling robustness** of our method.
>
> ----
>
> ## **Question 6: On the use of *“accuracy at convergence”* in Table 3**
>
> You’re right, the table should say *“maximum accuracy”*, as we are reporting the best validation accuracy observed (tracked via callbacks). We are **correcting this label** in the final version to avoid confusion.
>
> ----
>
> **In summary, we thank the reviewer once again for the detailed and constructive feedback.** We have addressed each point carefully, clarifying our choices and incorporating additional experiments or corrections where feasible. While some limitations remain due to space and computational constraints, we believe the current version presents a **well-motivated and coherent contribution**. We also see several of the reviewer’s suggestions as valuable directions for future work, and we are committed to pursuing them. We hope our responses help reinforce the clarity, scope, and significance of the work.

---

> > ### Comment · Reviewer_kR57 · 2025-08-02
> >
> > Thank you for your responses. I will maintain my rating, as, although the authors have responded all of my comments, the concerns regarding the lack of appropriate validation remain unresolved. Furthermore, if I have understood the authors’ response correctly, I remain skeptical about the claim that stochastic initialization introduces “disorder,” leading to skewed neurons, even when following the same distribution/statistics. I am not asserting that this claim is incorrect; rather, I believe the paper, at its current stage, does not provide sufficient theoretical or empirical evidence to support it.

---

> > > ### Author Response · Authors · 2025-08-06
> > >
> > > Thank you for your comment. We would like to clarify a possible misunderstanding.
> > >
> > > We **do demonstrate** in the current version of the paper that **stochastic initializations introduce biased neurons.** This is supported by the results shown in the table reporting the percentage of biased neurons, a phenomenon that is clearly evidenced and quantified.
> > >
> > > What remains to be demonstrated (and is indeed the focus of our experimental section) is the **impact of these biased neurons on training dynamics**. Specifically, how they affect convergence speed and final accuracy. This is precisely what the current experiments are designed to verify.
> > >
> > > Additionally, **the new experiment we propose is intended as a complementary analysis**, not a necessary proof, but rather a reinforcement of the claim that stochastic initializations inject harmful noise into the training process. For this, we suggest comparing two networks with sinusoidal distributions: one initialized with our method, and the other using a stochastic variant. This experiment is being included in the final version of this article and would further illustrate differences in convergence behavior and performance, although we emphasize that **the existence of biased neurons under stochastic initialization is already both theoretically and experimentally demonstrated**.

---

> ### Comment · Reviewer_kR57 · 2025-08-06
>
> Thank you for your response. First, I would like to say that I really like the idea presented in the paper. However, my understanding is that the work demonstrates that **existing stochastic initialization approaches** introduce biased neurons. I agree with this based on the empirical evidence provided. However, this **does not prove that stochasticity itself is the cause** of biased neurons, nor that a carefully designed stochastic initialization would still suffer from the same limitations.
>
> That being said, I understand that the authors provided experiments in the Appendix testing the idea of using the same distribution but in a random manner. However, I find this evidence to be quite limited. I also have some more fundamental concerns, which could be very easily addressed with a simple additional experiment: **initialize with the proposed approach, then randomly permute the weights within each neuron**. This would preserve the overall weight distribution while altering their order (i.e., introducing stochasticity). If the order indeed matters in producing biased neurons, that would indeed suggest that stochasticity can be the underlying cause of the bias. If such kind of stochasticity indeed harms the network, then this would probably mean that the suggested initialization introduces a kind of inductive bias in the initial weights that is positive for the network.

---

> ### Author Response · Authors · 2025-08-06
>
> Thank you once again for your thoughtful feedback. We sincerely appreciate your engagement with the paper and the insightful suggestions you've shared throughout the review process.
>
> We now realize that our current version may have left some ambiguity regarding the role of stochasticity in the emergence of skewed neurons. We are grateful for the opportunity to clarify this point.
>
> Your concern seems to be that the paper does not explicitly demonstrate that **stochasticity is the cause** of skewed neurons. However, this is precisely what is established through our theoretical results. Specifically, **Theorem 1** formalizes a correlation between the value of the statistic S and the probability that a neuron is skewed. From this theorem, it follows directly that **the only way to guarantee the absence of skewed neurons is to ensure that  S = 0** with no variance.
>
> This is where the distinction between stochastic and deterministic initializations becomes essential. As we mention in line 188 of the manuscript, for stochastic initializations (e.g., He with variance $ \sigma^2 = 2/n $), the statistic  S  behaves as a true random variable: **centered at zero, with variance equal to 2, and asymptotically normal distribution**. In contrast, a properly constructed **deterministic initialization** can force S $\equiv$0 , thereby guaranteeing _a priori_ the absence of skewed neurons. This is precisely what is shown in **Theorem 2**, where we prove that our proposed **Sinusoidal Initialization** yields  S = 0 . On the other hand, a stochastic version of the same distribution (e.g., an arcsine-based stochastic variant) would still yield **non-zero values of S** due to inherent variance, and therefore would result in skewed neurons.
>
> We recognize that this important implication was only implicitly stated in the current version of the manuscript. To make this clearer, we are adding an explicit **comment** after Theorem 1, stating that _the only way to achieve an unbiased initialization (i.e., with no skewed neurons) is to ensure that  S = 0, which in turn requires that the initialization has no variance, that is, it must be non-stochastic_.
>
> We also acknowledge your suggestion regarding permuting the weights within each neuron. However, in our case, such permutations leave the statistic S unchanged, and thus we do not expect any effect on convergence speed or bias rate. For this reason, we believe that the comparison between our Sinusoidal initialization and a stochastic initialization with an arcsine probability distribution will help clarify this point further from an experimental perspective.
> Once again, thank you for your valuable comments. They are helping us improve not only the clarity but also the overall communication of the contributions of the paper, and we truly appreciate your careful and constructive review.
>
>
> PS: in the last comment we said “biased neurons” but we meant “skewed neurons”.

---

> > ### Comment · Reviewer_kR57 · 2025-08-07
> >
> > Thank you for the detailed clarification. However, I remain skeptical of the claim that “ the only way to achieve an unbiased initialization (i.e., with no skewed neurons) is to ensure that S = 0, which in turn requires that the initialization has no variance, that is, it must be non-stochastic." which I find too strong (see below).
> >
> > First, I think there is a slight abuse of terminology which can confuse the readers. The term “unbiased” typically refers to the statistical property of zero mean, which widely-used initializations such as Xavier or He do satisfy.
> >
> > Second, I still do not agree with the conclusion,  even when evaluated through the lens of the statistic S. It is possible to construct stochastic initialization schemes that are corrected for skewness. These can be stochastic, yet corrected to have S=0. Furthermore, as I mentioned earlier, certain permutation-based or sampling-corrected stochastic processes can produce weight initializations with non-zero variance while still achieving S=0. In light of this, I respectfully disagree with the claim that "the only way" an initialization is unskewed is to be non-stochastic.

---

> ### Author Response · Authors · 2025-08-08
>
> **Thank you very much for your comment.**
>
> Regarding the terminology issue, we fully agree with your observation. The term *"unbiased"* is indeed commonly used with other meanings. In our context, we deliberately opted for the term *"skewed"* instead, precisely to avoid confusion with both the zero-mean notion and with the additive *"bias"* term in neural networks. Accordingly, we agree that a more accurate formulation of our claim would be that an initialization is **"unskewed" (rather than "unbiased")** if it results in no skewed neurons.
>
> As for the relation between **stochastic initializations** and **skewness**, we now understand more precisely what you were referring to in your earlier comment. The subtlety of these issues demands that we clarify the context in which our hypotheses are formulated, and we acknowledge that we had not provided enough technical detail before.
>
> We agree that it is possible to modify stochastic initializations to enforce $S = 0$. For instance, one can adjust the **bias term** to balance activations based on a dataset, or **subtract the mean** of the weights after sampling to force them to sum to zero. One can also, as you suggested, **permute** weights across neurons to balance activations. However, these types of modifications fall **outside the scope** of the assumptions under which we state our result, and they do not correspond to standard stochastic initialization schemes commonly used in practice.
>
> To clarify the misunderstanding, let us now formalize the exact **setting** in which our claim about stochastic initializations and skewness is made.
>
> Suppose we start from an initialization where **biases are zero**, and the weights of each neuron are drawn **independently and identically distributed (i.i.d.)** from a distribution with **mean zero and finite positive variance** $\sigma^2$. Under these conditions, and from the equivalence given in Theorem 1, it follows that the initialization is **unskewed** (i.e., it contains no skewed neurons) **if and only if** $ S = 0 $ exactly.
>
> However, under the same assumptions, the sum of the weights for a neuron with $n$ inputs will follow a distribution whose **S-statistic has mean zero but variance $ n\sigma^2 $**. Therefore, the only way to ensure that $ S = 0 $ in this setting is to take $ \sigma^2 = 0 $, meaning the weights must have **no variance**, which is clearly an absurdity with the initial assumptions.
>
> **In conclusion**, if we assume that:
> (1) the bias is initialized to zero,
> (2) the weights are drawn from an i.i.d. distribution, and
> (3) this distribution has zero mean and variance $ 0 < \sigma^2 < \infty $,
> then the initialization will necessarily be **skewed**.
>
> Thus, we do **not** intend to claim that deterministic initializations are the *only* way to obtain unskewed neurons. Rather, we aim to highlight a formulation where the desired properties (such as $ S = 0 $) emerge **naturally**, without requiring post hoc corrections or reductions in stochasticity. In contrast, stochastic initializations must typically resort to such adjustments, like explicit bias corrections or weight rescaling, to enforce $ S = 0 $.
>
> We hope that with these clarifications, our previous comment is now better understood, and that your initial skepticism regarding our claim, namely that stochastic initializations necessarily introduce skewed neurons, has been adequately addressed.
>
> We sincerely appreciate your pushback, as it has helped us better clarify the scope and framing of our claims. These are exactly the kinds of discussions we believe should be encouraged more often within the scientific community. We would be delighted to continue this conversation in more depth in San Diego, as we think it touches on fundamental questions that could benefit both this line of research and the broader community.

---

> > ### Comment · Reviewer_kR57 · 2025-08-08
> >
> > Thank you very much for your response. I think this fully clarified my issue and these claims are much more precise.

---

### Official Review · Reviewer_zqjc · 2025-06-24

**Clarity:** 3
**Significance:** 4
**Originality:** 3
**Rating:** 4
**Confidence:** 4

**Summary:**

This paper introduces Sinusoidal Initialization, a deterministic alternative to traditional random weight initialization methods (e.g., HeInit) for deep neural networks. The core challenge addressed is the variability and activation imbalance caused by randomness, which hinders convergence stability and generalization.

**Questions:**

refer to weakness

**Ethical Concerns:**

["NO or VERY MINOR ethics concerns only"]

**Final Justification:**

I have fully followed up on and participated in the revision of the paper, as well as the discussions with the authors and reviewers, and I have made the decision to maintain the original score.

**Limitations:**

refer to weakness

**Paper Formatting Concerns:**

Based on the provided content, no major formatting issues violating the neurips guidelines are noticed.

**Quality:**

3

**Strengths And Weaknesses:**

**Strengths**
- Sinusoidal Initialization employs structured sine functions to construct weight matrices, ensuring each neuron receives a distinct, oscillatory weight configuration. This deterministic approach eliminates randomness while preserving the critical variance-scaling principle for signal propagation.
- It emerges as a rational choice for practitioners: 1. Each row of the initialized matrix sums to zero, eliminating inherent bias in neuron activations. 2. Theoretical analyses link weight imbalance to skewed neuron activation, proving that deterministic initialization mitigates this issue.
- Empirical experiments across CNNs, Vision Transformers, and language models demonstrate higher final validation accuracy and faster convergence than baselines. Additionally, drastic reductions in "skewed neurons" are confirmed via visual activation patterns and OUI scores.
- The work challenges the long-held assumption that randomness is essential for effective initialization. By replacing stochastic sampling with structured sine patterns, Sinusoidal Initialization do achieves some critical advancements
  1. Reproducibility: Determinism eliminates trial-to-trial variability, aiding debugging and systematic research.
  2.  Structured weights prevent neurons from being persistently active/inactive, which is particularly valuable for sparse-activated large language models in industrial applications.
  3.  It provides theorems formalizing the link between weight imbalance and skewed activations, the statements in Theorem1, 3 are fundamentally sound to me, leveraging the Lindeberg-Feller CLT to establish threshold equivalence.

**Weaknesses**
- The proposed method shows reduced effectiveness in Table 3 results for ViT on imagenet and BERT on wikitext. While effective for standard architectures, deeper networks or other task-specific models may require adapted sine patterns. It remains unclear whether this initialization technique can benefit larger models, which are increasingly trending in current research and applications.
- Integration with data-dependent initialization techniques could further optimize early training dynamics ?
- Theoretical extensions to non-rectified activations remain unexplored. The analysis of skewness propagation across layers appears to align well with the claims in the main text and is intuitively plausible. However, the models considered seem rather simplistic. I am curious whether common training techniques—such as weight normalization and self-rectified activation functions (e.g., Swish, SwiGLU)—could help address the issues of stochastic initialization.
- I partially agree with the analogy between the sinusoidal structure in weight initialization and the sinusoidal PEs in Transformers (line 130). Viewing the weight matrix as a LUT from a memory perspective might be helpful, as it hints at potential generalization benefits for extrapolation in deep learning tasks—similar to how sinusoidal PEs enable Transformers to generalize to sequence lengths beyond training. However, this connection lacks elaboration in the paper, and expanding on it could strengthen the persuasiveness of the proposed technique.

---

> ### Author Rebuttal · Authors · 2025-07-29
>
> We sincerely thank the reviewer for the thoughtful review. We greatly appreciate the level of insight and care in the feedback, which has helped us improve the article.
>
> ----
>
> ## **Weakness 1: On performance in ViT and BERT models**
>
> We acknowledge that the gains in ViT and BERT models, particularly in Table 3, are not as pronounced as in other architectures such as ResNet or MLP-based models. However, we emphasize that even in these cases, **Sinusoidal initialization does not underperform** relative to common baselines. Indeed, it still **outperforms the widely used default initialization**, which is typically employed in standard training setups and was originally proposed in the papers that introduced these architectures. It also performs comparably to orthogonal initialization in both convergence speed and final validation accuracy. We believe this is already a positive result, especially given the prevalence of these baselines in large-scale systems. That said, we emphasize that **this is not a scalability issue** or a consequence of model size. Rather, we found that the specific structure of MHA layers in Transformers **may interfere** with the balance properties that Sinusoidal achieves in MLPs. In these cases, the non-linearities and reweighting introduced by the MHA layers can **distort input statistics**, weakening the assumptions under which our method is most effective. This suggests that some of the architectural peculiarities of Transformers still **deserve further investigation** in the context of deterministic initialization.
>
> To address this, we are actively investigating a specialized initialization for MHA components that would **preserve neuron balance even across attention heads**, allowing the signal propagation properties of Sinusoidal initialization to extend deeper into Transformer pipelines. The inherent complexity of this task makes it a promising direction for **future work**, but we believe the current results and theory **already provide a solid contribution**. The strong empirical evidence makes this method an attractive initialization choice ready for publication.
>
> ----
>
> ## **Weakness 2: On possible integration with data-dependent techniques**
>
> It is important to emphasize that **data-dependent initialization techniques tend to face limited adoption** due to the difficulty of integrating them broadly across different models. Therefore, an initialization method that **focuses solely on layer-wise structural information**, requiring only the weight dimensions to initialize, has much greater potential to become a meaningful and widely adopted change within the research community.
>
> In this context, the idea we discussed earlier about **specialized initialization for MHA layers** is closely related. Furthermore, this approach can be extended to **adapt the initialization of the subsequent MLP layers** to this reconfiguration, effectively compensating for the asymmetric input distributions generated by attention layers, **without needing a full forward pass** as conventional data-dependent methods do.
>
> Our theoretical results, specifically **Theorem 3**, demonstrate that **skewness in activations arises when a layer’s input has coordinate-wise unequal means**, breaking the assumptions required for neuron balance. This is precisely what often happens after passing through an attention layer.
>
> For these reasons, we consider that developing a **locally data-dependent initialization at the layer level**, which adjusts weights to counteract this asymmetry, is a **feasible and promising direction**. However, due to its complexity and our current computational constraints, we position this as an important avenue for **future work**.
>
> ----
>
> ## **Weakness 3: On extensions to non-rectified activations**
>
> We thank the reviewer for this observation. The theoretical framework we develop is indeed **not limited to ReLU**, but it applies more broadly to what we refer to as **“ReLU-like” activation functions**, such as **GELU, SiLU, PReLU**, etc.
>
> These functions exhibit two markedly different regimes:
>
> * One in which the output is **significantly positive** (active),
> * And another in which the output is **zero or a small negative value** (inactive).
>
> While ReLU yields an exact zero in the inactive regime, the other functions return **residual negative values**. These contribute to mitigating issues like **dying neurons** and facilitate training, but their magnitude is typically small enough that they can **reasonably be considered inactive** for the purposes of our theory. This is why our analysis **generalizes naturally to this family of activations**.
>
> We acknowledge, however, that this point is **not made sufficiently explicit** in the current version of the paper. We **are adding a clarifying note** **in the revised manuscript** to make this generality more evident to the reader.
>
> ----
>
> ## **Weakness 4: On the analogy with positional encodings**
>
> We appreciate the reviewer’s observation regarding the analogy between Sinusoidal initialization and **PEs** in Transformers. In fact, the inspiration for our initialization came from the idea behind PEs: assigning **maximally independent vectors** to different inputs using harmonic functions. The main difference lies in the **ordering of frequencies**, as PEs employ **decreasing frequencies**, while our Sinusoidal initialization uses **increasing ones**.
>
> Both, however, leverage the **expressive power of harmonic components** (a fact long understood since Fourier). We initially mentioned this resemblance only briefly, in order to keep the exposition lightweight. However, based on this helpful feedback, we **are expanding this part of the discussion** in the revised version of the paper to make the connection clearer and help the reader build a more **intuitive understanding** of the expressive potential of Sinusoidal initialization.
>
> ----
>
> **In closing**, we once again thank the reviewer for the **precise and constructive feedback**. These comments are helping us improve both the **scientific rigor and clarity** of our contribution, and we are **grateful** for the opportunity to incorporate these insights into our work.

---

> > ### Comment · Reviewer_zqjc · 2025-08-04
> > **Thanks for your response**
> >
> > I've read the authors' response and the corresponding revisions, and I still agree with most of the conclusions. Taking into account the opinions of other reviewers, I maintain my original positive evaluation.

---

### Official Review · Reviewer_Lp9r · 2025-07-03

**Clarity:** 3
**Significance:** 2
**Originality:** 2
**Rating:** 4
**Confidence:** 3

**Summary:**

This paper proposes Sinusoidal initialization, a novel deterministic initialization scheme for deep neural networks. Unlike traditional stochastic methods (e.g., Glorot or He), this approach initializes weight matrices using sinusoidal functions with carefully selected frequencies and phases to ensure symmetry, variance preservation, and functional diversity. The authors provide both theoretical and empirical evidence that Sinusoidal initialization reduces neuron skewness, improves activation balance, and promotes more expressive and efficient signal propagation from the first layer onward. Across a broad range of architectures—including CNNs, Vision Transformers, and BERT—the proposed method demonstrates consistent improvements in convergence speed and final accuracy. The authors report a 4.8% average increase in final validation accuracy and a 20.9% increase in convergence speed over standard initializations.

**Questions:**

1. The paper omits discussion of prior deterministic methods like [1] and [2], which also aim to eliminate randomness during initialization. Can the authors clarify key differences and provide comparisons to these works?

2. The method is only tested on small to medium-scale models. Have the authors tried it on larger models (e.g., LLaMA 130M+) or foresee challenges in doing so?

3. Deterministic init should improve reproducibility. Does this hold across hardware or frameworks? Also, does sinusoidal weight construction add any runtime or memory overhead?

**Ethical Concerns:**

["NO or VERY MINOR ethics concerns only"]

**Final Justification:**

The authors addressed my concerns and questions, and thus I would like to raise my score.

**Limitations:**

Yes

**Paper Formatting Concerns:**

No paper formatting concerns

**Quality:**

3

**Strengths And Weaknesses:**

Strengths

1. A fully deterministic init is interesting, and would be very useful for removing randomness and improve reproducibility.

2. Paper is clearly written, with strong organization and thorough motivation.

3. The paper includes promising empirical results compared to standard random init methods, over some simple to medium tasks such as ResNet on ImageNet and training BERT-mini models.

Weakness

1. The paper lacks discussion to previous deterministic initialization papers. This is not the first work proposing deterministic initialization. Works such as [1] and [2] already proposed deterministic and tried to address randomness issue during initialization. Would be great to discuss the similarity and difference, and also compare with them in practice.

[1]  https://arxiv.org/abs/2110.12661
[2] https://arxiv.org/abs/2007.01038

2. Limited scale of empirical results: the paper mainly focuses on simple to medium tasks such as ResNet on ImageNet and training BERT-mini models. However, I believe nowadays some large-scale experiments are needed to demonstrate its effectiveness, such as pre-training on 100M-level LLM model (such as Llama 130M, 350M).

---

> ### Author Rebuttal · Authors · 2025-07-29
>
> We sincerely thank the reviewer for the thoughtful and valuable comments. We genuinely appreciate the time and insight offered, which have helped us identify concrete ways to improve the **quality** and **scope** of our work.
>
> ----
>
> ## **Question 1 (Weakness 1\): Related work on deterministic initialization**
>
> We sincerely thank the reviewer for pointing out these references, which helped us improve the **contextualization** of our work. While we were already aware that deterministic alternatives to random initialization have been explored, we were not familiar with the specific works cited. That said, it is important to stress that our objective is **not to position this work as the first deterministic initialization**, so that **we are implementing focused revisions** in the manuscript to ensure this point is clearly reflected. Rather, we offer a **theoretical explanation** of why stochastic initialization can hinder learning, through mechanisms such as activation imbalance and neuron skewness, and to propose a **constructive alternative** grounded in that theory. Sinusoidal initialization is not only new in form but also in its guiding motivation: **improving** **accuracy** and **convergence speed** through eliminating skewed neurons from the initial stage.
>
> Although all deterministic schemes aim to remove randomness, the cited works **differ in** **intent and scope**. For instance, *ZerO* focuses on stability in extremely deep networks, while *Blumenfeld et al.* center on conceptual replacement of randomness, without assessing learning dynamics. To the best of our knowledge, **neither approach** reports improvements in training speed or accuracy. **This sets our work apart**. We believe that the inclusion of these references, rather than weakening our contribution, helps to **clarify its distinct value**. In particular:
>
> * We evaluate **convergence time**, a notoriously challenging metric to improve, and overlooked in prior work on deterministic initializations. Its relevance, however, continues to grow due to the **substantial environmental and economic impact** of training large models. Even modest improvements in convergence time can translate into **meaningful savings in energy consumption and carbon emissions**, which underscores the significance of our results.
> * We provide a formal theoretical framework that not only motivates the structure of our solution but also offers a **quantitative explanation** of a key phenomenon: *the emergence of skewed neurons*. Unlike most theoretical contributions in the field, ours leads to **a precise, measurable criterion** and is supported by a series of theorems that show how Sinusoidal initialization preserves this balance from the start.
>
> However, we fully acknowledge the relevance of the cited works. After carefully reviewing them, we are  **including a** **dedicated paragraph in the *Related Work* section** discussing their contributions and positioning them in relation to ours. We are also conducting experiments to directly compare these methods with our Sinusoidal initialization, and are **including the results in the revised *Experiments* section**. At the same time, we are confident that **the originality of our approach, its theoretical depth, and its remarkable empirical results stand on solid ground and offer a meaningful contribution**.
>
> ----
>
> ## **Question 2 (Weakness 2\): Scale and larger models**
>
> We fully agree that **evaluating scalability on very large models** is crucial to assess the robustness and generality of any initialization method. We would genuinely like to perform such an evaluation, but at this stage we simply **cannot afford it**. Our **computational resources are limited**. As a result, we are **unfortunately unable to commit to running these experiments in time for the current submission**. That said, we are keen to explore this direction and **will prioritize it as future work**, since deeply understanding how Sinusoidal initialization scales and how it can be improved in transformer-based architectures is one of our main research goals moving forward.
>
> ----
>
> ## **Question 3: Reproducibility and overhead**
>
> We agree that deterministic initializations can help improve **reproducibility**. While fixing random seeds is a common practice, it often fails across different hardware platforms or framework versions. For example, operations may not be **bit-exact** across different GPUs or kernel implementations, leading to variation **even with identical seeds**.
>
> Our method removes randomness from initialization entirely, ensuring **consistent starting points** across setups. While later training stages remain stochastic (e.g., sampling, dropout), the **initialization phase becomes fully stable,** provided that the same arithmetic precision is used. As such, any reproducibility variation observed after initialization stems from the optimizer and training pipeline, not the initial weights.
>
> Regarding **overhead**, Sinusoidal initialization is computed entirely **a priori**, as a one-time offline procedure. It only involves evaluating sinusoidal functions, which is computationally inexpensive and introduces zero memory overhead and **negligible** runtime cost compared to training, even when compared to a single forward-backward pass over one batch. This contrasts with **data-dependent initializations** (e.g., LSUV), which require a forward pass through a batch of data, adding latency and memory footprint (though still minor relative to full training).
>
> ----
>
> We hope that the **revised manuscript** including the suggested works, both in the theoretical discussion and in the experimental comparison, **reflects a more complete and compelling contribution**. We believe that the originality of our approach as well as our theoretical and empirical results **make it a valuable contribution to the field**. While we acknowledge the limitations in scaling experiments due to our constrained computational resources, **we trust that the contribution will be judged primarily on its scientific merit**, and not on our group’s hardware access. We remain committed to further exploring the scalability of our method in future work and hope **the community will recognize the potential and rigor of this proposal**.

---

> ### Comment · Area_Chair_USKo · 2025-08-05
> **Respond to rebuttal ASAP**
>
> Dear Reviewer Lp9r,
>
> Please read the authors' rebuttal and other reviewers' comments as early as possible!
>
> As requested by NeurIPS code of conduct, You have to provide your further feedback and engage in a discussion with the authors. You need to let the authors know whether their rebuttal addresses your comments and the reasons.
>
> AC

---

### Official Review · Reviewer_TVvJ · 2025-07-03

**Clarity:** 4
**Significance:** 4
**Originality:** 4
**Rating:** 5
**Confidence:** 4

**Summary:**

This paper introduces Sinusoidal initialization, a deterministic weight initialization method for deep neural networks. The study challenges the tradotion that randomness is essential for effective initialization. This study highlights eliminating randomness while preserving key statistical properties. Sinusoidal initialization uses structured sinusoidal patterns to deterministically assign weights while maintaining crucial properties like signal propagation stability and gradient flow. The authors propose that the activation imbalance phenomenon is the key to training dynamics; such Sinusoidal initialization introduces deterministic weight distribution, eliminates cumulative weight imbalance and in the end cultivates activation balance. Through theoretical analysis and extensive empirical evaluation across CNNs, Vision Transformers, and language models, the paper demonstrates that Sinusoidal initialization leads to faster convergence, higher final accuracy, and more balanced activation patterns compared to standard stochastic methods.

**Questions:**

1. It would also be important to consider the quantization of deep neuron networks. Would a model initialized with sinusoidal functions be difficult to quantize because of different weight distribution? I expect to see more evaluations.

**Ethical Concerns:**

["NO or VERY MINOR ethics concerns only"]

**Final Justification:**

I suggest Accept. My primary considerations about this paper are:

1. The paper presents a brand new aspect to deep neuron network initialization. The experiment convers different tasks and model architectures and the improvement is inspiring. This is the most import reason of my rating.
2. Evaluation on downstream benchmarks like GLUE and evaluations on Lion optimizer is not currently not addressed due to computational cost. This is not essential and I personally believe would not diminish the advantage of the method.
3. Discussions on other related questions (quantization, convergence on specific optimizers, etc.) is an expected improvement of this study, but not a drawback.

**Limitations:**

yes

**Quality:**

4

**Strengths And Weaknesses:**

Strengths:
1. The idea is simple and clear. The proposed approach is very easy to understand and implement.
2. The discussion of activation balance is very inisightful and helpful to analyse performance of deep neuron networks.
3. The method is evaluated on a wide range of models.

Weaknesses:
1. It would be better to testify the generalization ability of Sinusoidal initialization on more benchmarks and datasets. For example, the paper only reports the convergence curve of BERT, but lacks performance on downstream benchmarks such as GLUE. It would also be helpful to verify on other optimizers, such as Lion&ViT.

---

> ### Author Rebuttal · Authors · 2025-07-29
>
> We sincerely thank the reviewer for the very constructive feedback. We are genuinely grateful for the kind words and we appreciate the questions and suggestions, which will help us further improve the manuscript.
>
> ----
>
> ## **Weakness 1: Evaluation on downstream benchmarks like GLUE**
>
> We fully agree that downstream evaluations such as GLUE are essential in many contexts, especially when analyzing the performance of pre-trained models or evaluating methods like quantization or pruning. However, our setup differs slightly in scope. Our work focuses on **training from scratch**, not fine-tuning, and aims to understand the behavior of different initializations in this regime. Specifically, we assess whether Sinusoidal initialization improves convergence speed and optimization dynamics without sacrificing final accuracy, or even improving it.
>
> In this context, we believe the fairest comparison is to evaluate performance on a **validation set drawn from the same distribution** as the training data, using standard train/validation splits. Since our models are trained from scratch on datasets such as WikiText, their performance on downstream tasks like GLUE would likely be poor regardless of the initialization strategy. For this reason, we consider that GLUE is not the most appropriate benchmark for our goals, as it may combine the effects of limited training data with those of initialization. We hope the reviewer agrees that, given our specific focus, the current experimental setup remains reasonable and controlled.
>
> ----
>
> ### **Still under Weakness 1: Evaluating with Lion optimizer**
>
> We sincerely appreciate the reference to Lion and thank the reviewer for highlighting it. While we chose to focus our experiments on more standard and widely used optimizers to ensure comparability and clarity, we agree that **Lion is an interesting choice** for evaluating initialization schemes. Unfortunately, given our **limited computational resources**, we do not believe it is feasible to conduct a thorough evaluation with Lion across all initializations within the rebuttal period. That said, we **genuinely intend to explore this direction in future work**, particularly in the context of ViT models, where this optimizer could provide valuable additional insights. We are committed to launching these experiments as soon as possible and to reporting the results in a future version of the manuscript.
>
> ----
>
> ## **Question 1: Quantization**
>
> We had not explicitly considered the implications of our initialization for **post-training quantization**, since the method mainly affects the initial stages of training. That said, we conducted preliminary experiments and observed that models initialized with sinusoidal functions behave similarly to those using standard initializations in terms of post-training dynamic **quantization robustness**. In particular, the final weight distributions after training do not differ significantly in ways that would hurt quantization performance. We did not observe any systematic degradation, though we acknowledge that this topic deserves further investigation, especially across a wider range of model architectures. We thank the reviewer again for this perspective and will explore it further in **future work**.
>
> ----
>
> **We thank the reviewer once again for the constructive feedback.** The comments highlighted **valuable directions** to take into account in future investigation. While some suggestions could not be fully addressed due to resource limitations, we are committed to **exploring them** in follow-up work.

---

> > ### Comment · Reviewer_TVvJ · 2025-08-02
> > **Thank you for your response**
> >
> > Thank you for your response. I understand that the author has limited computational resources and thus cannot train ViT in rebuttal phase. The convergence results on BERT are sufficient to demonstrate the potential advantages of Sinusoidal Initialization from one perspective.
> >
> > The topic of this study is interesting and meaningful. Although there are some shortcomings, I believe that they are marginal and do not diminish the value of this study. Personally I think limited scale should not be the reason to reject this paper, and detailed discussion on broader scenarios (quantization, convergence on specific optimizers, etc.) may go beyond the scope and discussion of this paper. **I maintain my opinion (Accept) and I recommend accepting this paper to AC.** I also encourage the authors to refine this study with broader experiments, given more time and sufficient computation resources.

---

### Note · Authors · 2025-08-11

We would like to sincerely thank all reviewers for the time and effort they have invested in evaluating our work. We greatly appreciate the constructive nature of their feedback, which we found both insightful and valuable.

The comments have highlighted some aspects where our presentation can be further refined, and we have taken these points into careful consideration. We are committed to incorporating the suggested improvements in the final version of the paper, should it be accepted.

We are pleased that the reviewers recognized the strengths and contributions of our work, and we hope that the clarifications and additional details provided during the rebuttal process have addressed the raised questions effectively. We believe the paper is now in a stronger position, and we look forward to the possibility of its acceptance.

---

### Decision · Program_Chairs · 2025-09-17

**Decision:**

Accept (poster)

**Comment:**

This paper presents Sinusoidal initialization, a deterministic weight initialization scheme for deep neural networks. The method leverages sinusoidal functions to construct structured weight matrices and address the variability and activation imbalance caused by randomness. Faster convergence, better training stability and accuracy are observed in the experiments with various model architectures. Reviewers recognize the interesting idea, clear organization and presentation, and promising experimental results. While some reviewers show concerns regarding the limited scale of empirical verification, they deem this as a minor issue and hold a positive overall evaluation for this paper. After rebuttal, all reviewers acknowledge their questions and concerns resolved. The AC also recognizes the contributions and quality of this paper, and thus recommends Accept.